# Fourier-enhanced Implicit Neural Fusion Network for Multispectral and Hyperspectral Image Fusion

**Yu-Jie Liang**[†]
University of Electronic
Science and Technology of China
yujieliang0219@gmail.com

**Zihan Cao**[†]
University of Electronic
Science and Technology of China
iamzihan666@gmail.com

**Shangqi Deng**
Xi'an Jiaotong University
shangqideng0124@gmail.com

**Hong-Xia Dou**
Xihua University
hongxiadou1991@126.com

**Liang-Jian Deng**[*]
University of Electronic Science and Technology of China
liangjian.deng@uestc.edu.cn

## Abstract

Recently, implicit neural representations (INR) have made significant strides in various vision-related domains, providing a novel solution for Multispectral and Hyperspectral Image Fusion (MHIF) tasks. However, INR is prone to losing high-frequency information and is confined to the lack of global perceptual capabilities. To address these issues, this paper introduces a Fourier-enhanced Implicit Neural Fusion Network (FeINFN) specifically designed for MHIF task, targeting the following phenomena: *The Fourier amplitudes of the HR-HSI latent code and LR-HSI are remarkably similar; however, their phases exhibit different patterns.* In FeINFN, we innovatively propose a spatial and frequency implicit fusion function (Spa-Fre IFF), helping INR capture high-frequency information and expanding the receptive field. Besides, a new decoder employing a complex Gabor wavelet activation function, called Spatial-Frequency Interactive Decoder (SFID), is invented to enhance the interaction of INR features. Especially, we further theoretically prove that the Gabor wavelet activation possesses a time-frequency tightness property that favors learning the optimal bandwidths in the decoder. Experiments on two benchmark MHIF datasets verify the state-of-the-art (SOTA) performance of the proposed method, both visually and quantitatively. Also, ablation studies demonstrate the mentioned contributions. The code can be available at https://github.com/294coder/Efficient-MIF.

## 1 Introduction

Hyperspectral imaging captures scenes across contiguous spectral bands, offering intricate details compared to traditional single or limited-band images, and improving computer vision application accuracy, such as target recognition, classification [47], tracking, and segmentation [12, 38, 39, 37, 44, 48, 49]. However, practical optical sensors face challenges in balancing spatial resolution and spectral precision. Images with over 100 bands often exhibit lower spatial resolution, while those with fewer

---

[†]Equal contribution.
[*]Corresponding author.

38th Conference on Neural Information Processing Systems (NeurIPS 2024).

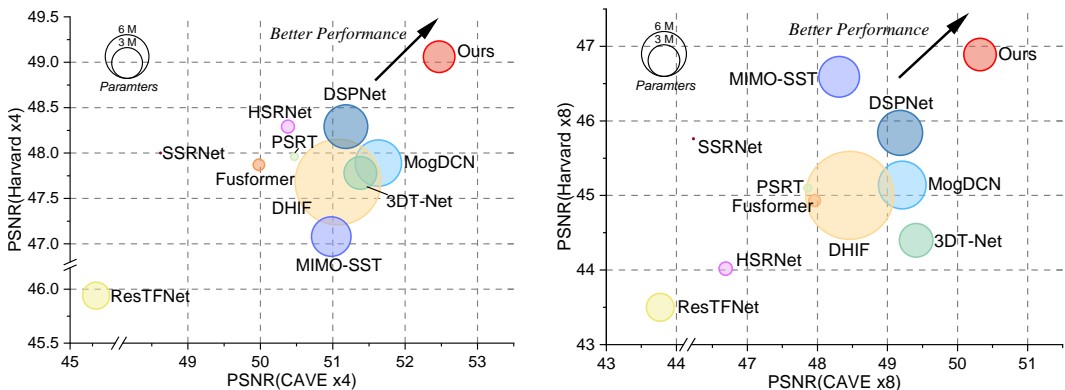

Figure 1: Comparison of our method with other methods on the CAVE($\times$ 4, $\times$ 8) and Harvard($\times$ 4, $\times$ 8) datasets. Closer to the top-right corner indicates better performance and the size of the circle indicates the number of parameters in the model.

bands display higher spatial resolution. Efforts for MHIF are underway to fuse high spatial-resolution multispectral images (HR-MSI) with low spatial-resolution hyperspectral images (LR-HSI) to finally obtain high spatial-resolution hyperspectral images (HR-HSI). Actually, MHIF technology could fuse hyperspectral images with multispectral images, extracting information not detectable by HR-MSI to enhance richness and precision. Recent MHIF literature explores model-based approaches [8, 9, 45] and deep learning methods [17, 10, 3, 54]. While model-based methods leverage image priors, challenges persist in obtaining high-fidelity, low-distortion HR-HSI due to the lack of large-scale training datasets. Among deep-learning approaches, CNN-based networks for HR-MSI and LR-HSI tend to be limited and lack interpretability for MHIF tasks and Transformer frameworks [15, 7] address the small receptive field of CNN but bring greater computational overhead.

In recent years, implicit representations of 3D scenes have garnered significant attention from researchers. For instance, Neural Radiance Field [43] models 3D static scenes by mapping coordinates to signals through a neural network. Inspired by this, researchers have revisited image representation for 2D tasks. Recent studies [5, 20, 34, 4] have achieved arbitrary-scale super-resolution (SR) by replacing commonly used upsampling layers with local implicit image functions. Though these methods demonstrate superior performance in 2D tasks, they still have some drawbacks. *Firstly*, INR calculates the RGB values of a queried coordinate based on the relative distances to the surrounding four pixels, treating it as a local operation in space that lacks consideration for global information. *Additionally*, the MLP-ReLU structure used in traditional INR inherent high-frequency information bias [29] which is challenging to be eliminated during training.

To address these issues, we propose implicit fusion functions tailored for the MHIF task as a novel fusion paradigm. We first employ encoders to extract prior information from LR-HSI and HR-MSI, which is then fed into the implicit fusion functions in the form of latent codes. Unlike traditional INR, we transform latent codes into the Fourier domain and simultaneously perform spatial and frequency fusion in a unified network. This approach not only rectifies the high-frequency insensitivity induced by the MLP but also effectively extends the receptive field, encompassing a more comprehensive scope of global information. To integrate spatial and frequency domain representations efficiently, we design a decoder with time-frequency tightness, mapping features on both domains to pixel space. The contributions of this work are three folds:

- We define a novel fusion framework based on INR, which innovatively extracts information from the spatial and Fourier domains, effectively enhances the representation ability of high-frequency information, and expands the receptive field.

- We propose a new decoder employing a Gabor wavelet activation function to enhance the interaction of INR features. Furthermore, we theoretically prove that the complex Gabor wavelet activation possesses a time-frequency tightness property, which facilitates the decoder in learning the optimal bandwidths.

- The proposed network reaches state-of-the-art (SOTA) performance on the MHIF task across two widely used hyperspectral datasets at various fusion ratios. Fig. 1 provides a fair comparison with other SOTA methods.

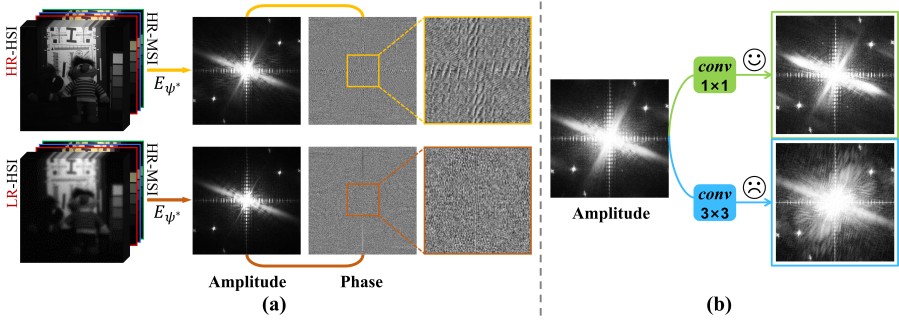

Figure 2: **(a)** The amplitude of latent code from the encoder fed by HR-HSI and LR-HSI (combined with HR-MSI) share a similarity, but the phases differ from each other. $E_{\psi*}$ is a trained encoder. **(b)** $3 \times 3$ convolution would suffer from the issue of spectrum leakage, which can be alleviated by $1 \times 1$ convolution.

## 2 Related works

**Implicit Neural Representation**    Unlike traditional discrete representations, neural implicit representation (INR) provides a more elegant and continuous parameterized approach. Initially applied in 3D modeling tasks, NeRF [43] revolutionized 3D computer vision by representing intricate three-dimensional scenes with just 2D pose images. This line of work extends to the 2D imaging domain, where INR performs a weighted average on adjacent sub-codes to ensure output value continuity. LIIF [5] recently introduces a local implicit image function for SR, leveraging MLP to sample pixel signals across the spatial domain. Several improvements focus on decoding networks; for example, UltraSR [46] incorporates residual networks, merging spatial coordinates and depth encoding. DIINN [26] utilizes a dual-interactive implicit neural network to decouple content and position features, improving decoding capabilities. JIIF [36] proposes joint implicit image functions for multimodal learning, extracting priors from guided images. Regarding activation functions in the MLP, SIREN [34] recommends utilizing periodic activation functions for continuous INR to fit complex signals. On the other hand, WIRE [32] further employs continuous complex Gabor wavelet activation functions to activate non-linearity, focusing more on spatial frequencies. However, there is limited research dedicated to designing INR architectures specifically for the MHIF task. The unique characteristics of hyperspectral images pose challenges for INR networks, in their insensitivity to high-frequency information.

**Latent Enhancement by Fourier Transform**    Fourier transform is a commonly used time-frequency analysis technique in signal processing, which converts signals from the time domain to the frequency domain. The Fourier domain has global statistical properties, and in recent years, many works use the Fourier transform to enhance the representation ability of neural networks. For example, FDA [53] proposes exchanging amplitude and phase components in Fourier space between images to enhance and adjust frequency information. FFC [6] introduces a novel convolution module that internally fuses cross-scale information to capture global features in Fourier space. Similarly, GFNet [30] uses 2D discrete Fourier transform to extract features, implements learnable global filtering, and replaces the self-attention layer in Transformer. UHDFour [21] embeds Fourier transform into the image enhancement network to model global information. Together, these studies demonstrate the utility of frequency domain information in improving performance on visual tasks. We exploit the architecture of FeINFN to transform latent codes into the frequency domain, implicitly integrating representations of amplitude and phase components, and enhancing high-frequency injection.

**Motivation**    [29] finds that most neural networks exhibit a phenomenon of spectral bias through Fourier analysis. This includes neural networks such as MLP, which tend to learn low-frequency information during the early stages of training and are insensitive to high-frequency information. Moreover, we found this issue occurs in the MHIF task according to an experimental analysis as shown in Fig. 2(a), where HR-HSI and LR-HSI were concatenated with HR-MSI and fed into a trained encoder to obtain latent codes. These codes were transformed into the frequency domain to visualize the amplitude and phase. It can be observed that the amplitudes from HR-HSI and LR-HSI are very similar, while the phases differ significantly. The phase of HR-HSI should naturally contain more texture than LR-HSI, a hypothesis validated by the visualized phase maps. Based on this finding, we transformed the latent codes into the Fourier domain to separately process amplitude and phase, to enhance the global learning of high-frequency information in the images.

# 3 Methodology

In this section, we first present the preliminary of INR and then provide the proposed framework tailored for MHIF task. Subsequently, we elaborate on the implementation details of the composited modules of the proposed FeINFN.

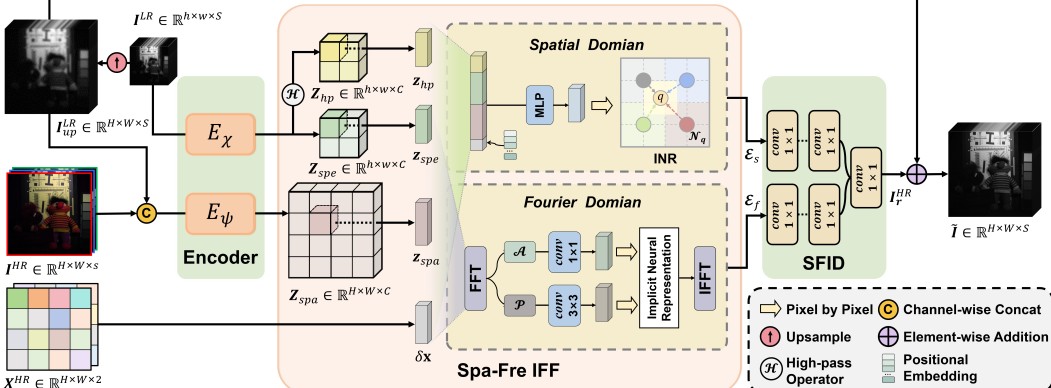

Figure 3: The flowchart of the FeINFN framework which is composed of a spectral encoder $E_\chi$, a spatial encoder $E_\psi$, MHIF task-designed spatial and Fourier domains implicit fusion functions, and a pixel space mapping decoder. Please note that $\mathbf{I}^{LR}$ is the LR-HSI, $\mathbf{I}^{HR}$ is the HR-MSI, $\mathbf{I}^{LR}_{up}$ is the bicubic interpolation LR-HSI, and $\mathbf{X}^{HR}$ is the HR normalized 2D coordinate map. $\mathbf{z}_{spe}$, $\mathbf{z}_{spa}$, $\mathbf{z}_{hp}$, $\delta\mathbf{x}$ correspond to individual pixel units, $\mathcal{A}$ and $\mathcal{P}$ represents amplitude and phase, respectively.

## 3.1 Preliminary: Implicit Neural Representation

Neural Radiance Fields [43] is represented by integral construction scenes. The value of a pixel in a certain viewing angle image is regarded as the integral of the characteristics of the sampling point from the proximal end to the far end of the ray. During actual training, the integral needs to be discretized. Extended to 2D image representation [5], it is sampled pixel by pixel from the vicinity of the query target. Taking the low-resolution (LR) image $\mathbf{I} \in \mathbb{R}^{h \times w \times 3}$ upsampling to the high-resolution (HR) image $\hat{\mathbf{I}} \in \mathbb{R}^{H \times W \times 3}$ as an example, the process of generating the RGB values of the target coordinates $\mathbf{x}_q \in \mathbb{R}^2$ can be regarded as interpolation form, expressed as:

$$\hat{\mathbf{I}}(\mathbf{x}_q) = \sum_{i \in \mathcal{N}_q} w_{q,i} \mathbf{v}_{q,i}, \tag{1}$$

where $\mathbf{v}_{q,i} \in \mathbb{R}^{4 \times 4 \times 3}$ is the interpolation pixel of $i$ interpolated by $q$'s surrounding pixels $\mathcal{N}_q \in \mathbb{R}^4$ and $w_{q,i} \in \mathbb{R}$ signifies the interpolation weight. In the implicit representation of local image features, the weights $w_{q,i} = S_i/S$, where $S_i$ represents the area formed by $q$ and $i$ in the diagonal region and $S$ denotes the total area enclosed by the set $\mathcal{N}_q$. The interpolation value $\mathbf{v}_{q,i}$ is effectively generated by a basis function:

$$\mathbf{v}_{q,i} = \phi_\theta(\mathbf{z}_i, \mathbf{x}_q - \mathbf{x}_i), \tag{2}$$

where $\phi_\theta$ is typically an MLP, $\mathbf{z}_i$ is the latent code generated by an encoder for the coordinates $\mathbf{x}_i$, and $\mathbf{x}_q - \mathbf{x}_i$ represents the relative coordinates. From the above equations, it can be inferred that the interpolation features can be represented by a set of local feature vectors in the LR domain. Typically, interpolation-based methods [28, 18] achieve upsampling by querying $\mathbf{x}_q - \mathbf{x}_i$ in the arbitrary SR task. See more details in [5].

## 3.2 Overview of the FeINFN Framework

In this work, we propose the FeINFN, which adopts a novel framework for simultaneously performing neural implicit representation in both the spatial and frequency domains to execute the MHIF task. Fig. 3 provides an overview of the proposed framework, designed to fuse LR-HSI $\mathbf{I}^{LR} \in \mathbb{R}^{h \times w \times S}$ and HR-MSI $\mathbf{I}^{HR} \in \mathbb{R}^{H \times W \times s}$ to generate HR-HSI $\widetilde{\mathbf{I}} \in \mathbb{R}^{H \times W \times S}$ based on a upsampling scale $r$.

Initially, the LR-HSI is fed into encoder $E_\chi$ to extract spectral features $\mathbf{Z}_{spe} \in \mathbb{R}^{h \times w \times C}$. Simultaneously, the concatenated bicubic interpolation LR-HSI $\mathbf{I}_{up}^{LR} \in \mathbb{R}^{H \times W \times S}$ and $\mathbf{I}^{HR}$, are fed into encoder $E_\psi$ to extract spatial features $\mathbf{Z}_{spa} \in \mathbb{R}^{H \times W \times C}$. Additionally, the pixel's central position is represented as the coordinate point. The coordinate map is normalized into a two-dimensional grid $[-1, 1] \times [-1, 1]$, obtaining a HR normalized 2D coordinate map $\mathbf{X}^{HR} \in \mathbb{R}^{H \times W \times 2}$. The extracted $\mathbf{Z}_{spe}$ and $\mathbf{Z}_{spa}$, along with the 2D coordinates of $\mathbf{I}^{HR}$, are forwarded to Spatial-Frequency Implicit Fusion Function (Spa-Fre IFF), outputting spatial domain features $\mathcal{E}_s \in \mathbb{R}^{H \times W \times S}$ and frequency domain features $\mathcal{E}_f \in \mathbb{R}^{H \times W \times S}$. The $\mathcal{E}_s$ and $\mathcal{E}_f$ as inputs to a pixel space mapping decoder which generates the residual image $\mathbf{I}_r^{HR} \in \mathbb{R}^{H \times W \times S}$. Finally, the residual image $\mathbf{I}_r^{HR}$ is combined with the bicubicly upsampled image $\mathbf{I}_{up}^{LR}$ via element-wise addition, yielding the ultimate fusion image $\widetilde{\mathbf{I}}$.

### 3.3 INR Encoder Networks

Analogous to local implicit representation functions [5, 20, 34, 4], the initial step involves extracting latent code representations. For the MHIF task, we address the challenges of both upsampling and fusion simultaneously, employing implicit neural representations as the solution.

The INR encoders try to extract spatial and spectral latent codes $\mathbf{Z}_{spa} \in \mathbb{R}^{H \times W \times C}, \mathbf{Z}_{spe} \in \mathbb{R}^{h \times w \times C}$: one is extracted from $\mathbf{I}^{LR}$, serving as the carrier for spectral information; the other is encoded from the concatenation of $\mathbf{I}_{up}^{LR}$ and $\mathbf{I}^{HR}$, aiding in spatial information during the fusion process. This process can be denoted as:

$$\mathbf{Z}_{spe} = E_\chi(\mathbf{I}^{LR}), \quad \mathbf{Z}_{spa} = E_\psi\left(\mathrm{Cat}(\mathbf{I}_{up}^{LR}, \mathbf{I}^{HR})\right), \tag{3}$$

where $E_\chi$ is the spectral encoder parameterized by $\chi$, $E_\psi$ is the spatial encoder parameterized by $\psi$, and $\mathrm{Cat}(\mathbf{I}_{up}^{LR}, \mathbf{I}^{HR})$ denotes the concatenation along the channel dimension. In practice, we utilize EDSR [23] as INR encoder networks.

### 3.4 Spatial-Frequency Implicit Fusion Function

To address the mentioned issues 2, we propose Spatial-Frequency Implicit Fusion Function, dubbed Spa-Fre IFF which is a dual-branch fusion function and utilized for computing the fusion feature of $\mathbf{Z}_{spe}$ and $\mathbf{Z}_{spa}$ in the spatial and frequency domains, respectively. Given a queried HR coordinate $\mathbf{x}_q \in \mathbf{X}^{HR}$ of a pixel unit $q$, Spa-Fre IFF estimates spatial feature vector $\boldsymbol{\varepsilon}_s \in \mathbb{R}^{1 \times 1 \times S}$ ($\boldsymbol{\varepsilon}_s \in \mathcal{E}_s$) and frequency feature vector $\boldsymbol{\varepsilon}_f \in \mathbb{R}^{1 \times 1 \times S}$ ($\boldsymbol{\varepsilon}_f \in \mathcal{E}_f$) as follows:

$$\boldsymbol{\varepsilon}_s, \boldsymbol{\varepsilon}_f = \text{Spa-Fre IFF}(\mathbf{z}_{spe}, \mathbf{z}_{spa}, \delta\mathbf{x}), \tag{4}$$

where $\mathbf{z}_{spe} \in \mathbb{R}^{1 \times 1 \times C}$ represents the spectral latent code vector corresponding to $\mathbf{x}_q$, and $\mathbf{z}_{spa} \in \mathbb{R}^{4 \times 4 \times C}$ is the spatial latent code vector. $\delta\mathbf{x}$ denotes the set of local relative coordinates, expressed by the following formula:

$$\delta\mathbf{x} = \{\mathbf{x}_q - \mathbf{x}_{q,i}\}_{i \in \mathcal{N}_q}, \tag{5}$$

where $\mathbf{x}_{q,i}$ refers to the coordinates most proximate to the query coordinate $\mathbf{x}_q$, representing the four corner pixels closest to $q$ in the HR space.

**Spatial Implicit Fusion Function** The Spatial Implicit Fusion Function aims to leverage the powerful representation capabilities of INR to achieve implicit fusion in the spatial domain, as shown in Fig. 3 (see branch "Spatial Domain"). Specifically, we employ high-pass operators $\mathcal{H}$ to filter the spectral latent codes, as a complement to the high-frequency information on the spectrum:

$$\mathbf{z}_{hp} = \mathcal{H}(\mathbf{z}_{spe}), \tag{6}$$

where $\mathbf{z}_{hp} \in \mathbb{R}^{1 \times 1 \times C}$ represents the high-frequency latent code of $\mathbf{I}^{LR}$. Also, we suggest frequency encoding for relative positional coordinates as follows:

$$\gamma(\delta\mathbf{x}) = [\sin(2^0 \delta\mathbf{x}), \cos(2^0 \delta\mathbf{x}), \cdots, \sin(2^{L-1} \delta\mathbf{x}), \cos(2^{L-1} \delta\mathbf{x})], \tag{7}$$

where $L$ is a hyperparameter, in practice, we set $L$ to 10. Additionally, leveraging the graph attention mechanism [36], we parameterize the solution for interpolation weights $\mathbf{w}_{q,i} \in \mathbb{R}^{1 \times S}$, and the

implicit fusion function simultaneously outputs fusion interpolation values $\mathbf{v}_{q,i} \in \mathbb{R}^{4 \times 4 \times S}$ and interpolation weights $\mathbf{w}_{q,i}$. The implicit fusion function is specifically expressed as:

$$\mathbf{w}_{q,i}, \mathbf{v}_{q,i} = \phi_\theta(\mathbf{z}_{spe}, \mathbf{z}_{spa}, \mathbf{z}_{hp}, \gamma(\delta \mathbf{x})), \tag{8}$$

where $\phi_\theta$ is an MLP parameterized by $\theta$. The weights used for interpolation need to pass through a softmax function, obtaining normalized weights $\overline{\mathbf{w}}_{q,i}$. The spatial implicit fusion interpolation, as shown in Eq. (1), yields the fused spatial feature $\boldsymbol{\varepsilon}_s \in \mathbb{R}^{1 \times 1 \times S}$ and can be described as follows:

$$\boldsymbol{\varepsilon}_s = \sum_{i \in \mathcal{N}_q} \overline{\mathbf{w}}_{q,i} * \mathbf{v}_{q,i}. \tag{9}$$

**Frequency Implicit Fusion Function**    From Fig. 2(a), we observed characteristics in the frequency features between LR-HSI and HR-HSI. Hence, we design a frequency implicit fusion function to express global features continuously in the Fourier domain. Notably, directly applying static kernel convolution in the frequency domain would only enhance a specific frequency range, which is inappropriate for the fusion task. However, by learning feature content to generate weights, INR can be seen as a dynamic interpolation method in continuous space, adaptively enhancing information in the frequency domain without overly altering the frequency distribution. Therefore, introducing INR into the Fourier domain is reasonable. Since amplitude and phase exhibit different forms, as shown in Fig. 2(a), we handle them separately.

With the considerations mentioned above, as illustrated in Fig. 3 (see branch "Fourier Domain"), we initially employ FFT to transform latent codes $\mathbf{z}_{spe}$ and $\mathbf{z}_{spa}$ from the spatial domain to the frequency domain, obtaining $\mathbf{f}_{spe} \in \mathbb{R}^{1 \times 1 \times C}$ and $\mathbf{f}_{spa} \in \mathbb{R}^{4 \times 4 \times C}$. After the transformation, we further obtain amplitude components $\mathcal{A}(\mathbf{f}_{spe})$ and $\mathcal{A}(\mathbf{f}_{spa})$, as well as phase components $\mathcal{P}(\mathbf{f}_{spe})$ and $\mathcal{P}(\mathbf{f}_{spa})$.

*For the amplitude*, as shown in Fig. 2(b), the amplitude distribution of LR-HSI and HR-HSI are very similar, and the non-point-wise convolution (*e.g.* Conv $3 \times 3$) causes an issue of spectrum leakage, confusing channel information. In contrast, point-wise convolution does not span multiple locations in the frequency domain and has no overlap allowing it to capture information across channels effectively. Thus the fusion function for amplitude components is more suitable when applying point-wise convolution:

$$\mathbf{w}_{q,i}^{\mathcal{A}}, \mathbf{v}_{q,i}^{\mathcal{A}} = \phi_\alpha^{\mathcal{A}}(\mathcal{A}(\mathbf{f}_{spe}), \mathcal{A}(\mathbf{f}_{spa}), \delta \mathbf{x}), \tag{10}$$

where $\mathbf{w}_{q,i}^{\mathcal{A}} \in \mathbb{R}^{1 \times S}$ and $\mathbf{v}_{q,i}^{\mathcal{A}} \in \mathbb{R}^{4 \times 4 \times S}$ are the weights and interpolated values for the corresponding amplitude component, and $\phi_\alpha^{\mathcal{A}}$ is a simple network composed of two layers of point convolutions parameterized by $\alpha$. Similar to operations in the spatial domain, implicit fusion interpolation is performed after obtaining interpolated values $\mathbf{v}_{q,i}^{\mathcal{A}}$ and the normalized weights $\overline{\mathbf{w}}_{q,i}^{\mathcal{A}}$:

$$\mathcal{A}_f' = \sum_{i \in \mathcal{N}_q} \overline{\mathbf{w}}_{q,i}^{\mathcal{A}} * \mathbf{v}_{q,i}^{\mathcal{A}}, \tag{11}$$

where $\mathcal{A}_f' \in \mathbb{R}^{1 \times 1 \times S}$ is the integrated amplitude component.

*For the phase*, which encapsulates information such as texture details, LR-HSI and HR-HSI often have different phase information. It is known that point convolutions fail to capture sufficient spatial representations. Therefore, we use a $3 \times 3$ convolution to learn phase information. Additionally, small changes in the frequency domain may result in significant variations in the spatial domain. We still consider using the form of INR interpolation for phase learning. The handling of the phase components $\mathcal{P}(\mathbf{f}_{spe})$ and $\mathcal{P}(\mathbf{f}_{spa})$ are formally similar to Eqs. (10) and (11):

$$\mathbf{w}_{q,i}^{\mathcal{P}}, \mathbf{v}_{q,i}^{\mathcal{P}} = \phi_\beta^{\mathcal{P}}(\mathcal{P}(\mathbf{f}_{spe}), \mathcal{P}(\mathbf{f}_{spa}), \delta(\mathbf{x})), \quad \mathcal{P}_f' = \sum_{i \in \mathcal{N}_q} \overline{\mathbf{w}}_{q,i}^{\mathcal{P}} * \mathbf{v}_{q,i}^{\mathcal{P}}. \tag{12}$$

The simple network $\phi_\beta^{\mathcal{P}}$ consists of two layers of $3 \times 3$ convolutions parameterized by $\beta$. $\mathcal{P}_f' \in \mathbb{R}^{1 \times 1 \times S}$ represents the integrated phase component.

Finally, IFFT is applied to map the frequency features $\mathcal{A}_f'$ and $\mathcal{P}_f'$ back to the image space, obtaining the frequency domain feature $\varepsilon_f \in \mathcal{E}_f$. Since in frequency space, one frequency point may correspond to multiple pixels at different positions in the spatial domain, the receptive field of INR in the frequency domain is enlarged in the spatial domain.

## 3.5 Spatial-Frequency Interactive Decoder

After obtaining the spatial feature map and frequency domain feature map, it is essential to consider how to integrate them seamlessly. Firstly, our decoder needs to have dual input and interactive capabilities. Secondly, it is necessary to focus on representing images in the spatial-frequency domain. With this in mind, we introduce the complex Gabor wavelet activation function with good time-frequency tightness and propose the Spatial-Frequency Interactive Decoder (SFID). Specifically, SFID consists

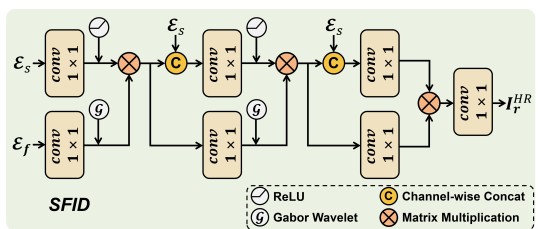

Figure 4: Detailed composition of the proposed SFID.

of three layers, taking spatial and frequency domain features as inputs. The outputs $\mathbf{I}_r^{HR}$ and $\mathbf{I}_{up}^{HR}$ contribute to the final fused image $\widetilde{\mathbf{I}}$. The decoding process is illustrated in Fig. 4. The complex Gabor wavelet function is defined as:

$$\mathcal{G}(\mathbf{x}) = e^{j\omega_0 \mathbf{x}} e^{-|v_0 \mathbf{x}|^2}, \tag{13}$$

where $\omega_0$ is the center frequency in the frequency domain, $v_0$ is a constant that is considered as the standard deviation of the Gaussian function, and $\mathbf{x}$ is a vector in the time (or spatial) domain. In what follows, we provide a theorem below that this Gabor wavelet activation has time-frequency tightness [1], which is helpful for the decoder's information interaction.

**Theorem 1.** *The complex Gabor wavelet activation in Eq.* (13) *has the time-frequency tightness property (more preliminary can be found in [1]). Moreover, from the perspective of signal spectrum analysis, this activation helps the decoder learn the optimal bandwidths.*

*Proof:* First, for the time-domas, the function $|\mathcal{G}(\mathbf{x})|$ in the time domain is primarily concentrated around $\mathbf{x} = 0$ due to the exponential decay term. The Gaussian term $e^{-|v_0 \mathbf{x}|^2}$ ensures that $\mathcal{G}(\mathbf{x})$ is bounded and rapidly decreases in the time domain. Second, for the frequency-domain Tightness, the Fourier transform is given by:

$$\mathcal{F}[\mathcal{G}(\mathbf{x})] = \int e^{j\omega_0 \mathbf{x}} e^{-|v_0 \mathbf{x}|^2} e^{-j\omega \mathbf{x}} d\mathbf{x}. \tag{14}$$

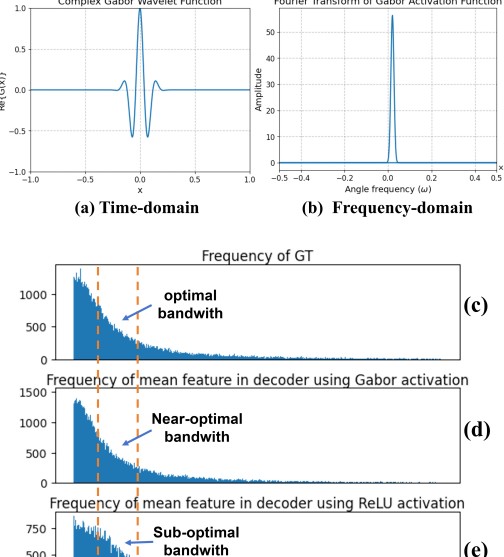

The Fourier transform of the Gaussian term $e^{-|v_0 \mathbf{x}|^2}$ remains a Gaussian function, and its bandwidth in the frequency domain is influenced by $v_0$. Due to the characteristics of the Gaussian function in the frequency domain, $\mathcal{G}(\mathbf{x})$ is mainly concentrated around $\omega = \omega_0$. Combining the narrow-bandwidth properties in both time and frequency domains, we can apply the uncertainty principle to demonstrate the time-frequency tightness of the complex Gabor function:

$$|\omega_0| \cdot v_0 \geq \frac{1}{4\pi}, \tag{15}$$

where $v_0$ is the time-domain bandwidth, and $|\omega_0|$ is the frequency-domain bandwidth. In practical training, we provide an initial set of bandwidths and allow the network to learn the optimal bandwidths, which concludes the proof. $\square$

Figure 5: **The complex Gabor wavelet function.** (a) and (b) depict the visualization of the complex Gabor wavelet function. (c), (d), and (e) represent the frequency responses of GT and decoder's mean feature using Gabor and regular ReLU activations, respectively.

As depicted in Fig. 5, the frequency response of the decoder with Gabor wavelet activation closely approximates the optimal bandwidth. Moreover, the decoder with Gabor activation achieves consistency with GT in frequency, demonstrating rapid frequency alignment.

Table 1: The average and standard deviation calculated for all the compared approaches on 11 CAVE examples and 10 Harvard examples simulating a scaling factor of 4. The best results are in bold, second-best in underline. "M" refers to millions.

| Methods | CAVE ×4 | | | | | Harvard ×4 | | | | |
|---|---|---|---|---|---|---|---|---|---|---|
| | PSNR(↑) | SAM(↓) | ERGAS(↓) | SSIM(↑) | #params | PSNR(↑) | SAM(↓) | ERGAS(↓) | SSIM(↑) | #params |
| Bicubic | 34.33±3.88 | 4.45±1.62 | 7.21±4.90 | 0.944±0.029 | – | 38.71±4.33 | 2.53±0.67 | 4.45±1.81 | 0.948±0.027 | – |
| CSTF-FUS [22] | 34.46±4.28 | 14.37±5.30 | 8.29±5.29 | 0.866±0.075 | – | 39.15±3.45 | 6.93±2.69 | 4.66±1.81 | 0.914±0.049 | – |
| LTTR [9] | 35.85±3.49 | 6.99±2.55 | 5.99±2.92 | 0.956±0.029 | – | 40.88±3.94 | 4.01±1.27 | 4.03±2.18 | 0.957±0.035 | – |
| LTMR [8] | 36.54±3.30 | 6.71±2.19 | 5.39±2.53 | 0.963±0.021 | – | 42.06±3.56 | 3.51±0.99 | 3.59±2.03 | 0.970±0.020 | – |
| IR-TenSR [45] | 35.61±3.45 | 12.30±4.68 | 5.90±3.05 | 0.945±0.027 | – | 40.47±3.04 | 4.36±1.52 | 5.57±1.57 | 0.963±0.014 | – |
| ResTFNet [24] | 45.58±5.47 | 2.82±0.70 | 2.36±2.59 | 0.993±0.006 | 2.387M | 45.94±4.35 | 2.61±0.69 | 2.56±1.32 | 0.985±0.008 | 2.387M |
| SSRNet [52] | 48.62±3.92 | 2.54±0.84 | 1.63±1.21 | 0.995±0.002 | **0.027M** | 48.00±3.36 | 2.31±0.60 | 2.30±1.42 | 0.987±0.007 | **0.027M** |
| HSRNet [16] | 50.38±3.38 | 2.23±0.66 | 1.20±0.75 | 0.996±0.001 | 0.633M | 48.29±3.03 | 2.26±0.56 | 1.87±0.81 | 0.988±0.006 | 0.633M |
| MogDCN [10] | 51.63±4.10 | 2.03±0.62 | 1.11±0.82 | 0.997±0.002 | 6.840M | 47.89±4.09 | 2.11±0.52 | 1.89±0.82 | 0.988±0.007 | 6.840M |
| Fusformer [15] | 49.98±8.10 | 2.20±0.85 | 2.50±5.21 | 0.994±0.011 | 0.504M | 47.87±5.13 | 2.84±2.07 | 2.04±0.99 | 0.986±0.010 | 0.467M |
| DHIF [17] | 51.07±4.17 | 2.01±0.63 | 1.22±0.97 | 0.997±0.002 | 22.462M | 47.68±3.85 | 2.32±0.53 | 1.95±0.92 | 0.988±0.007 | 22.462M |
| PSRT [7] | 50.47±6.19 | 2.19±0.64 | 2.06±3.71 | 0.996±0.003 | 0.247M | 47.96±3.21 | 2.18±0.55 | 1.89±0.86 | 0.988±0.006 | 0.247M |
| 3DT-Net [25] | 51.38±4.18 | 2.16±0.70 | 1.14±1.00 | 0.996±0.003 | 3.464M | 47.78±4.42 | **2.04±0.51** | 1.98±0.86 | **0.989±0.006** | 3.464M |
| DSPNet [35] | 51.18±3.92 | 2.15±0.64 | 1.13±0.82 | 0.997±0.002 | 6.064M | 48.29±3.16 | 2.30±0.55 | 1.93±0.93 | 0.988±0.006 | 6.064M |
| MIMO-SST [11] | 50.98±3.39 | 2.23±0.70 | 1.18±0.73 | 0.997±0.002 | 4.983M | 47.08±5.56 | 2.09±0.53 | 2.07±0.82 | 0.988±0.007 | 4.983M |
| FeINFN(Ours) | **52.47±4.10** | **1.91±0.59** | **0.98±0.74** | **0.998±0.002** | 3.165M | **49.06±3.15** | 2.10±0.53 | **1.78±0.75** | 0.989±0.007 | 3.165M |

# 4 Experiments

**Datasets** To evaluate the efficacy of our model, we conducted experiments using the CAVE and Harvard datasets. The CAVE dataset comprises 32 Hyperspectral Images (HSIs) with 31 spectral bands spanning from 400 nm to 700 nm at 10 nm intervals. We randomly selected 20 images for training and used the remaining 11 for testing. The Harvard dataset consists of 77 HSIs depicting indoor and outdoor scenes, covering the spectral range from 420 nm to 720 nm. We standardized the data by cropping the upper left sections of 20 Harvard images, with 10 for training and the rest for testing. The simulation of data can be found in Appendix.

**Implementation Details** We implement the proposed method FeINFN with Pytorch [27] on a workstation with an Intel I9 CPU and two 3090 GPUs. The optimizer is chosen as AdamW [19] and we use a Cosine anneal learning rate scheduler. The base channel number of the encoder is 128, that of the proposed implicit fusion function is 32 and in the decoder, the channel number is 31.

**Results on CAVE Dataset** In this section, we evaluate the effectiveness of FeINFN on the CAVE dataset and compare it with five traditional methods and some state-of-the-art deep learning-based approaches. As shown in Tab. 1 on the left, our method achieves optimal performance in the tasks of ×4 in all metrics. In the ×4 experiment, compared to currently leading methods such as DSPNet [35], 3DT-Net [25], and MogDCN [10], our approach demonstrates improvements in PSNR by 1.29dB/1.09dB/0.84dB, respectively. Notably, our method exhibits even more pronounced superiority in the ×8 experiment, showcasing good generalization across various resolutions. To illustrate the advantages of our method, we provide visual comparisons in Fig. 6, including close-ups and error maps to highlight specific details. Our fusion results closely match the ground truth, achieving the best quality. In comparing error maps, the darker colors indicate closer proximity to the original image. In contrast to other excellent methods, the error maps of FeINFN distinctly exhibit superior restoration effects on details.

**Results on Harvard Dataset** In Tab. 1, the right columns present the comparison results of our FeINFN with other methods on the Harvard dataset at scale factors 4. Our method performs exceptionally well, with only SAM being slightly surpassed by 3DT-Net [25]. FeINFN exhibits significant gains in PSNR/ERGAS metrics compared to the current state-of-the-art [16], with improvements of 0.77dB/0.09, respectively. The results with a scale factor of 8 can be found in Appendix. As depicted in Fig. 1, our model outperforms others, highlighting the crucial role of FeINFN's continuous representation capability in high-scale factor scenarios. To better visualize the performance gap, Fig. 6 illustrates the fused images and error maps, confirming that our FeINFN maintains high fidelity in recovering the texture details of the images.

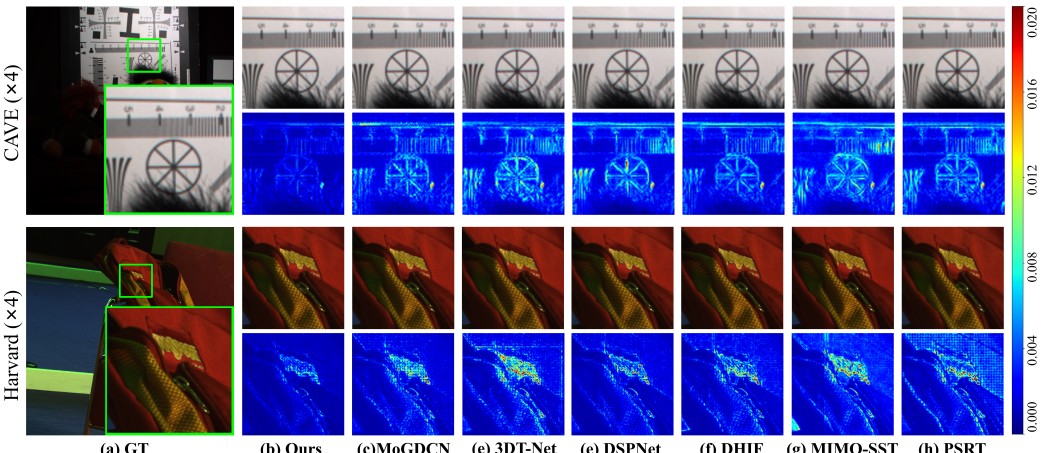

**(a) GT**    **(b) Ours**    **(c)MoGDCN**    **(e) 3DT-Net**    **(e) DSPNet**    **(f) DHIF**    **(g) MIMO-SST**    **(h) PSRT**

Figure 6: The upper and lower parts respectively showcase the results of "*Chart and Stuffed Toy*" from the CAVE dataset and "*Backpack*" from the Harvard dataset using pseudo-color representation. Green rectangles depict some close-up shots. The second and fourth rows show the residuals between the ground truth (GT) and the fusion products.

Table 2: Quantitative comparisons with other up-sampling methods on the CAVE (×4) dataset.

| Methods | PSNR(↑) | SAM(↓) | ERGAS(↓) | SSIM(↑) | #params |
|---|---|---|---|---|---|
| Bilinear | 52.23±4.40 | 1.92±0.60 | 1.03±0.86 | 0.997±0.0021 | 3.119M |
| Bicubic | 52.22±4.31 | 1.95±0.61 | 1.02±0.82 | 0.997±0.0021 | 3.119M |
| Pixel Shuffle | 52.26±4.37 | **1.90±0.59** | 1.02±0.85 | 0.997±0.0022 | 3.057M |
| Our | **52.47±4.10** | 1.91±0.59 | **0.98±0.74** | **0.998±0.0015** | 3.165M |

Table 3: Quantitative comparisons with reduced models on the CAVE (×4) dataset. $\mathcal{S}$ & $\mathcal{F}$ mean the domain difference.

| $\mathcal{S}$ | $\mathcal{F}$ | PSNR(↑) | SAM(↓) | ERGAS(↓) | SSIM(↑) | #params |
|---|---|---|---|---|---|---|
| ✓ | ✗ | 52.11±4.22 | 1.95±0.59 | 1.04±0.82 | 0.998±0.0017 | 2.869M |
| ✗ | ✓ | 47.86±3.42 | 3.49±1.30 | 1.67±1.13 | 0.995±0.0020 | 2.940M |
| ✓ | ✓ | **52.47±4.10** | **1.91±0.59** | **0.98±0.74** | **0.998±0.0015** | 3.165M |

Table 4: Quantitative comparisons with different activation functions in SFID on the CAVE (×4) dataset.

| Nonlinear | PSNR(↑) | SAM(↓) | ERGAS(↓) | SSIM(↑) |
|---|---|---|---|---|
| ReLU | 52.03±3.84 | 2.00±0.59 | 1.02±0.74 | **0.998±0.0013** |
| GELU | 51.96±3.88 | 2.01±0.60 | 1.03±0.75 | 0.998±0.0014 |
| Leaky ReLU | 51.98±3.92 | 2.01±0.60 | 1.03±0.76 | 0.998±0.0014 |
| Our | **52.47±4.10** | **1.91±0.59** | **0.98±0.74** | 0.998±0.0015 |

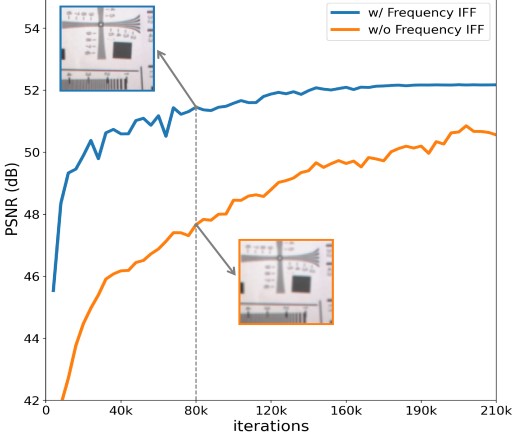

Figure 7: Changes in PSNR on the CAVE dataset of our FeINFN over iterations with and without the "Fourier Domain". The Frequency IFF can help the network learn the high-frequency details and converge faster.

## 4.1 Ablation Studies

**Upsampling Methods** Implicit image representation can be seen as an advanced interpolation algorithm, offering additional spatial information and parameterized weight generation. In this section, we compare INR with other upsampling methods. We replace INR with pixel-shuffle [33] and traditional CNN interpolation methods, presenting a comparative analysis. As seen in Tab. 2, our approach outperforms other methods in MHIF tasks.

**Spatial Domain and Fourier Domain** To assess the dual-domain model's efficacy, we performed model reduction, preserving spatial and Fourier domains independently. As shown in Tab. 3, FeINFN excels by using both spatial and Fourier domains concurrently, underscoring the positive impact of Fourier domain integration on overall network performance.

Spectral deviation occurs during training, where the network tends to prioritize low-frequency information, capturing high-frequency details only in later stages. To validate our resolution of this issue, we remove the "Fourier Domain" from Spa-Fre IFF, or retain it, and the corresponding training

data is illustrated in Fig. 7. Our FeINFN, which incorporates Fourier domain fusion, leads to faster PSNR convergence and overall higher efficiency. The visual comparison of high-frequency details in "*chart and stuffed toy*" from the cave dataset at 80k iterations further supports the significant improvement achieved with our results.

**Decoder with Different Nonlinear**    In this section, we evaluate the impact of different activation functions in SFID, aiming to match SFIFF. Our dual-input decoder incorporates a complex Gabor wavelet activation function to facilitate the fusion of spatial and frequency domain features.

Through experiments, we replaced the Gabor wavelet activation with other activations, presenting the results in Tab. 4. The findings distinctly demonstrate the enhanced fusion quality achieved with the complex Gabor wavelet activation. This emphasizes the critical role of wavelet activation in promoting robust and reliable learning in SFID.

## 5    Conclusion

Inspired by the distinct behaviors of LR-HSI and HR-HSI in the Fourier domain, we introduce a novel Fourier-enhanced Implicit Neural Fusion Network (FeINFN) based on INR. Through Fourier transformation, latent features are converted into the frequency domain, allowing the modeling of frequency components to enrich high-frequency information in images. Additionally, we propose a spatial-frequency decoding module, achieving a unified representation of both spatial and frequency domains using a time-frequency-tight activation function. Thanks to the unique design of our network, it outperforms state-of-the-art methods in MHIF with appealing efficiency. We desire that our work will inspire future research on frequency fusion-based MHIF methods.

## 6    Acknowledgement

This work is supported by the National Natural Science Foundation of China under Grants 12271083, 12171072 and Natural Science Foundation of Sichuan Province under Grants 2023NSFSC1341.

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

# A  Appendix / supplemental material

This supplementary material provides additional insights into the background, methodologies, and experimental details outlined in our paper. It includes limitations and broader impact, experiments compute resources, details on CNNs in MHIF, 2D Fourier transform, an elucidation of the global receptive field of convolution operations within the Fourier Domain , a description of data simulation, and quality metrics. Furthermore, we present a comprehensive comparison of all methods applied to the CAVE and Harvard datasets with a scale factor of 8. This includes an ablation study, affirming the efficacy of incorporating the Fourier domain on the CAVE($\times 8$). The provided information aims to enhance the reader's understanding of the intricacies involved in our research and its practical applications.

## A.1  Limitations and Broader Impact

**Limitations**  This study has certain limitations that should be acknowledged. One primary limitation is the unavailability of ground truth (GT) data in real-world settings for the task of multispectral and hyperspectral image fusion. Due to this constraint, all datasets used in our experiments are simulated. The detailed steps for data simulation can be found in Appendix A.6. This reliance on simulated data may affect the generalizability of our results to real-world scenarios. Consequently, while our methods show promising performance in experiments, their effectiveness in practical applications remains to be fully validated. However, this limitation is a challenge faced by the entire field, not unique to our work. Surveys [40, 31] in the field of multispectral and hyperspectral image fusion highlight this common issue and discuss the need for improved data simulation methods and benchmarks.

**Broader Impact**  This research addresses the task of multispectral and hyperspectral image fusion, which is crucial for enhancing the spatial resolution of hyperspectral images while preserving their spectral fidelity. The resulting high-resolution hyperspectral images (HR-HSI) are invaluable for various applications, such as resource monitoring, environmental management, and urban planning. In the environmental domain, fused images aid in pollution tracking, vegetation analysis, and precision agriculture, contributing to sustainable practices and environmental protection. These fused images facilitate more accurate and detailed analysis in these fields, potentially leading to better-informed decisions and more effective resource management. Despite these benefits, there are potential negative consequences to consider. Image fusion is a low-level task that significantly impacts subsequent image-processing steps. If the fusion process fails, resulting in distorted HR-HSI, it may adversely affect follow-up tasks and analyses, leading to incorrect conclusions or misguided decisions. Thus, ensuring the robustness and accuracy of the image fusion algorithm is critical to mitigating these risks.

## A.2  Experiments Compute Resources

Our experiments were conducted on a workstation equipped with an Intel 12th Gen i7-12700K processor, two NVIDIA RTX 3090 GPUs, and 128GB of memory. This setup provided sufficient computational power to handle the intensive tasks involved in multispectral and hyperspectral image fusion.

## A.3  Related Works: CNNs in MHIF

In recent years, CNN-based methods have exhibited significant success in the domain of multispectral and hyperspectral image fusion (MHIF). Their efficacy lies in their adeptness to extract high-level features from input data through end-to-end learning. SSRNet [52] leverages three distinct convolutional modules, w.r.t, a fusion module, a spatial edge module, and a spectral edge module, excelling in image reconstruction by associating a spatial-spectral loss function, contributing to robust learning outcomes. Similarly, ResTFNet [24] adopts residual structures and a two-stream fusion network, drawing inspiration from the extensive application of ResNet [13] in super-resolution image processing. In contrast, the MHF network [50] incorporates a well-explored linear mapping that connects HR-HSI to HR-MSI and LR-HSI, facilitating ease of interpretation. MoG-DCN [10] employs a dedicated subnet to approximate the decomposition matrix and conducts hyperspectral image super-resolution using DCN-based image regularization, leveraging prior knowledge of HSI. For the

simultaneous extraction of spatial and spectral information and the acquisition of high-quality details, HSRNet [16] integrates spatial and channel attention modules, enhancing the fusion performance. However, due to the model scaling and limited convolutional receptive field, the CNN-based models still struggle to obtain satisfactory results for the MHIF task.

## A.4 Preliminary: 2D Fourier Transform

Fourier transform is commonly employed in digital signal processing [2], aiming to convert signals from the time domain to the frequency domain. Through this domain transformation, previously imperceptible features often become observable. For two-dimensional images, the Fourier transform converts the signal from the spatial domain to the frequency domain, enabling the transformation of images into spectrograms in the frequency domain. Given a single-channel image $\mathbf{X} \in \mathbb{R}^{H \times W}$, the Fourier transform translates it into the Fourier space as the complex component $\mathbf{Y} \in \mathbb{C}^{H \times W}$. This process can be represented as follows:

$$\mathcal{F}(\mathbf{X})(u, v) = \mathbf{Y}(u, v) = \frac{1}{\sqrt{HW}} \sum_{h=0}^{H-1} \sum_{w=0}^{W-1} \mathbf{X}(h, w) e^{-j2\pi \left( \frac{hu}{H} + \frac{wv}{W} \right)}, \tag{16}$$

where $(h, w)$ denotes the coordinates of $\mathbf{x}$ in the spatial space, and $(u, v)$ represent the coordinates of $\mathbf{Y}$ in the Fourier space. The Fourier space is spanned by complex orthogonal basis functions, and each complex frequency component can be expressed as amplitude $\mathcal{A}(\mathbf{Y}(u, v))$ and phase $\mathcal{P}(\mathbf{Y}(u, v))$ components:

$$\mathcal{A}(\mathbf{Y}(u, v)) = \sqrt{\Re^2 \{\mathbf{Y}(u, v)\} + \Im^2 \{\mathbf{Y}(u, v)\}}, \tag{17}$$

$$\mathcal{P}(\mathbf{Y}(u, v)) = \arctan \left[ \frac{\Im(\mathbf{Y}(u, v))}{\Re(\mathbf{Y}(u, v))} \right], \tag{18}$$

where $\Re(\mathbf{Y})$ and $\Im(\mathbf{Y})$ respectively represent the real and imaginary parts. For multi-channel images, in the utilization of the Fourier transform, we perform individual Fourier calculations for each channel. Additionally, the Fourier transform is a reversible transformation, enabling bidirectional conversion between the original signal and the transformed signal, we denote $\mathcal{F}^{-1}$ as the Fourier inverse transform.

## A.5 The Receptive Field of INR in the Fourier Domain

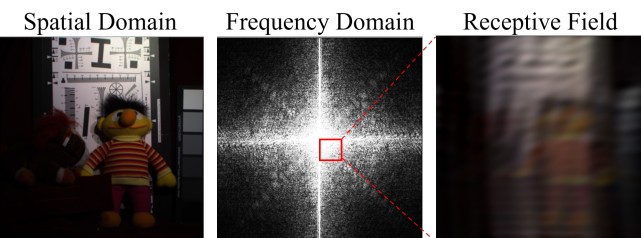

Figure 8: Convolving an image in the frequency domain is globally impactful in the spatial domain.

To validate the global nature of implicit feature fusion in the Fourier domain, we conducted experiments as illustrated in Fig. 8. We transform the "*chart and stuffed toy*" sample from the CAVE dataset into the Fourier domain, performed convolutions only on specific frequency features, and then transform it back to the spatial domain. It can be observed that Fourier domain convolution yields a global response in the spatial domain. Performing INR in the frequency domain indeed expands the receptive field of INR, freeing it from local constraints.

## A.6 Data Simulation

The proposed architecture takes LR-HSI and HR-MSI pairs $(\mathbf{I}^{LR}, \mathbf{I}^{HR})$ as input, with the training ground-truth (GT) being HR-HSI. However, due to the unavailability of HR-HSI as a reference, a simulation phase is necessary. In our experiments using the CAVE dataset, we cropped 20 training images, generating 3920 overlapping patches of size $64 \times 64 \times 31$. These patches serve as HR-HSI (ground truth) patches. To simulate appropriate LR-HSIs, we applied a $3 \times 3$ Gaussian blur kernel

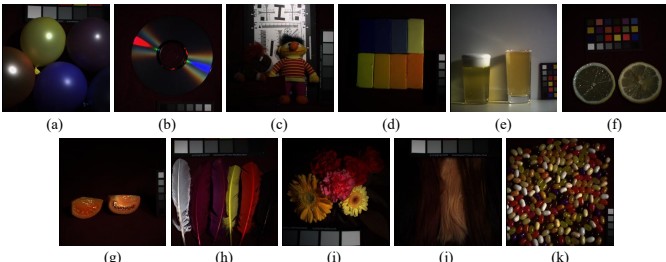

Figure 9: The testing images from the CAVE dataset: (a) *balloons*, (b) *cd*, (c) *chart and stuffed toy*, (d) *clay*, (e) *fake and real beers*, (f) *fake and real lemon slices*, (g) *fake and real tomatoes*, (h) *feathers*, (i) *flowers*, (j) *hairs*, and (k) *jelly beans*. An RGB color representation is used to depict the images.

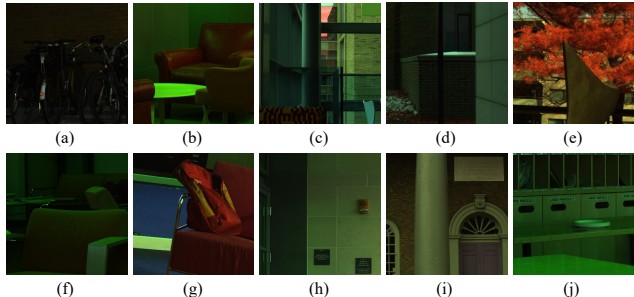

Figure 10: The 10 images tested on the Harvard dataset are (a) *bikes*, (b) *sofa1*, (c) *window*, (d) *fence*, (e) *tree*, (f) *sofa2*, (g) *backpack*, (h) *wall*, (i) *door* and (j) *parcels*.

with a standard deviation of $0.5$ to the original HR-HSIs. Subsequently, we downsampled the blurred patches by a factor of $4$. HR-MSI patches were generated using the spectral response function of a Nikon D700 camera. Therefore, input pairs ($\mathbf{I}^{LR}$, $\mathbf{I}^{HR}$) consist of 3920 LR-HSI patches of size $16 \times 16 \times 31$ and RGB image patches of size $64 \times 64 \times 3$. Paired with their corresponding GTs, these pairs were randomly split into training data ($80\%$) and validation data ($20\%$). The testing set of the CAVE dataset is shown in Fig. 9. The same procedure was employed to simulate the input LR-HSI and HR-MSI pairs and GTs for the Harvard dataset. The Harvard test set is shown in Fig. 10.

### A.7 Quality Metrics

We compare our method with other methods using different image quality metrics to validate the image fusion capability of our model, including the Spectral Angle Mapper (SAM) [51], the Erreur Relative Globale Adimensionnelle de Synthèse (ERGAS) [41], the Peak Signal-to-Noise Ratio (PSNR) [14], and the Structural SIMilarity (SSIM) [42].

PSNR evaluates the spatial quality of each band in the reconstructed HR-HSI. It is calculated as follows:

$$\text{PSNR}(\mathbf{I}, \widetilde{\mathbf{I}}) = \frac{1}{B} \sum_{i=1}^{B} \text{PSNR}(\mathbf{I}^i, \widetilde{\mathbf{I}}^i), \tag{19}$$

Here, $\mathbf{I}^i \in \mathbb{R}^{H \times W}$ and $\widetilde{\mathbf{I}}^i \in \mathbb{R}^{H \times W}$ represent the $i$-th band of $\mathbf{I} \in \mathbb{R}^{H \times W \times B}$ and $\widetilde{\mathbf{I}} \in \mathbb{R}^{H \times W \times B}$, respectively. The PSNR function is defined as:

$$\text{PSNR}(\mathbf{I}^i, \widetilde{\mathbf{I}}^i) = 20 \cdot \log_{10} \left( \frac{\max(\mathbf{I}^i)}{\sqrt{\text{MSE}(\mathbf{I}^i, \widetilde{\mathbf{I}}^i)}} \right), \tag{20}$$

where MSE (Mean Square Error) between $\mathbf{I}^i$ and $\widetilde{\mathbf{I}}^i$, and $\max(\cdot)$ is the maximum value of $\mathbf{I}^i$.

SAM measures the spectral distortion of each hyperspectral pixel in the reconstructed HR-HSI. It is given by:

$$\text{SAM}(\mathbf{I}, \widetilde{\mathbf{I}}) = \frac{1}{HW} \sum_{i=1}^{HW} \cos^{-1}\left(\frac{\mathbf{I}_i^T \widetilde{\mathbf{I}}_i}{||\mathbf{I}_i||_2 ||\widetilde{\mathbf{I}}_i||_2}\right), \tag{21}$$

where $\cos^{-1}$ is the arccosine function, $\mathbf{I}_i \in \mathbb{R}^{B \times 1}$ and $\widetilde{\mathbf{I}}_i \in \mathbb{R}^{B \times 1}$ are the spectra of the $i$-th pixel of $\mathbf{I}$ and $\widetilde{\mathbf{I}}$, respectively, $|| \cdot ||_2$ is the $\ell_2$ norm, and $\cdot^T$ denotes the transpose.

ERGAS measures the global statistical quality of the reconstructed HR-HSI, taking into account the ratio of the ground sample distances between HR-MSI and LR-HSI. It is formulated as:

$$\text{ERGAS}(\mathbf{I}, \widetilde{\mathbf{I}}) = \frac{100}{c} \sqrt{\frac{1}{B} \sum_{i=1}^{B} \frac{\text{MSE}(\mathbf{I}^i, \widetilde{\mathbf{I}}^i)}{\mu_{\widetilde{\mathbf{I}}^i}^2}}, \tag{22}$$

where $c$ is the scaling factor, and $\mu_{\widetilde{\mathbf{I}}^i}^2$ is the square of the mean value of $\widetilde{\mathbf{I}}^i$.

SSIM is used to assess the structural differences between GT and the reconstructed HR-HS, incorporating both luminance and structural contrast functions. The SSIM function is defined as:

$$\text{SSIM}(\mathbf{I}, \widetilde{\mathbf{I}}) = \frac{1}{B} \sum_{i=1}^{B} \frac{(2\mu_{\mathbf{I}^i}\mu_{\widetilde{\mathbf{I}}^i} + C_1)(2\sigma_{\mathbf{I}^i\widetilde{\mathbf{I}}^i} + C_2)}{(\mu_{\mathbf{I}^i}^2 + \mu_{\widetilde{\mathbf{I}}^i}^2 + C_1)(\sigma_{\mathbf{I}^i}^2 + \sigma_{\widetilde{\mathbf{I}}^i}^2 + C_2)}, \tag{23}$$

where, $B$ is the number of bands, and $\mathbf{I}$ and $\widetilde{\mathbf{I}}$ are sets containing $\mathbf{I}^i$ and $\widetilde{\mathbf{I}}^i$ for $i = 1$ to $B$, respectively. $\mu_{\mathbf{I}^i}$ and $\mu_{\widetilde{\mathbf{I}}^i}$ represent the mean values of $\mathbf{I}^i$ and $\widetilde{\mathbf{I}}^i$, while $\sigma_{\mathbf{I}^i}^2$ and $\sigma_{\widetilde{\mathbf{I}}^i}^2$ denote their variances. The term $\sigma_{\mathbf{I}^i\widetilde{\mathbf{I}}^i}$ indicates the covariance between $\mathbf{I}^i$ and $\widetilde{\mathbf{I}}^i$. Constants $C_1$ and $C_2$ are fixed values.

Higher PSNR values indicate better performance, while lower SAM and ERGAS values signify higher quality of the reconstructed HR-HSI. SSIM values range from $-1$ to $1$, with values closer to $1$ indicating better quality. Ideally, PSNR should be infinite, SAM and ERGAS should be zero, and SSIM should be one.

## A.8 Benchmark

To evaluate FeINFN's performance, we compare it with MHIF methods on the CAVE and Harvard datasets. The bicubic-interpolated result of the upsampled LR-HSI in Tab. 1 serves as our baseline. Various model-based techniques, including the CSTF-FUS [22], LTTR [9], LTMR [8], and IR-TenSR [45] approaches, are considered. Additionally, we compare our approach with various deep learning methods, such as SSRNet [52], ResTFNet [24], HSRNet [16], MoGDCN [10], Fusformer [15], and DHIF [17], PSRT [7], 3DT-Net [25], DSPNet [35], MIMO-SST [11]. We compare our method with other methods using different image quality metrics to validate the image fusion capability of our model, including SAM [51], ERGAS [41], PSNR [14], and SSIM [42].

## A.9 More Comparisons with the Larger Scaling Factor on CAVE and Harvard Datasets

Due to space constraints in the main text, we present a more detailed comparison of our FeINFN with other methods on four metrics for the CAVE dataset and the Harvard dataset in the supplementary material. As shown in Tab. 5, FeINFN demonstrates the best overall performance. While it ranks second in SAM on the CAVE dataset, it maintains the optimal results for other metrics.

## A.10 Ablation Study: The Effectiveness of Fourier Domain Incorporation on CAVE ($\times 8$)

In the main text, we described our ablation experiments to investigate the effectiveness of incorporating the Fourier domain. Additionally, we were interested in assessing its efficacy at larger scale factors. Therefore, we applied pruning to the model on the CAVE ($\times 8$) dataset to validate its effectiveness. The experimental results, as shown in Tab. 6, indicate that the model performs best when incorporating the Fourier domain operation, aligning with our hypothesis. This provides robust evidence supporting the enhancement of network performance through INR in the Fourier domain.

Table 5: The average and standard deviation calculated for all the compared approaches on 11 CAVE examples and 10 Harvard examples simulating a scaling factor of 8. The best results are in bold, second-best in underline. "M" refers to millions.

| Methods | CAVE ×8 | | | | | | Harvard ×8 | | | | | |
|---|---|---|---|---|---|---|---|---|---|---|---|---|
| | PSNR(↑) | SAM(↓) | ERGAS(↓) | SSIM(↑) | #params | #flops | PSNR(↑) | SAM(↓) | ERGAS(↓) | SSIM(↑) | #params | #flops |
| Bicubic | 29.96±3.54 | 5.89±2.32 | 5.56±3.08 | 0.887±0.066 | – | – | 33.18±6.85 | 3.10±0.90 | 3.83±1.84 | 0.894±0.073 | – | – |
| CSTF-FUS [22] | 38.44±4.25 | 7.00±2.65 | 2.11±1.15 | 0.959±0.033 | – | – | 39.84±6.51 | 4.49±1.52 | 2.40±1.84 | 0.932±0.092 | – | – |
| LTTR [9] | 37.92±3.59 | 5.37±1.96 | 2.44±1.05 | 0.972±0.018 | – | – | 42.09±4.56 | 3.62±1.34 | 1.80±0.96 | 0.960±0.048 | – | – |
| LTMR [8] | 38.41±3.57 | 5.04±1.70 | 2.24±0.97 | 0.974±0.017 | – | – | 42.09±4.56 | 3.62±1.34 | 1.80±0.92 | 0.959±0.060 | – | – |
| IR-TenSR [45] | 36.79±3.64 | 12.87±4.98 | 2.68±1.41 | 0.944±0.031 | – | – | 40.04±3.89 | 5.40±1.76 | 4.75±1.55 | 0.958±0.016 | – | – |
| ResTFNet [24] | 43.77±5.34 | 3.49±0.94 | 1.38±1.25 | 0.992±0.006 | 2.387M | 1.75G | 43.50±3.96 | 3.53±1.11 | 1.74±0.93 | 0.979±0.012 | 2.387M | 1.75G |
| SSRNet [52] | 46.23±4.19 | 3.13±0.97 | 1.05±0.73 | 0.993±0.004 | 0.446M | 0.11G | 45.76±3.34 | 2.99±0.98 | 1.34±0.74 | 0.983±0.010 | 0.027M | 0.11G |
| HSRNet [16] | 46.69±4.48 | 2.91±0.86 | 0.93±0.63 | 0.994±0.003 | 3.010M | 2.00G | 44.02±4.89 | 3.64±1.79 | 1.49±0.81 | 0.980±0.013 | 0.633M | 2.00G |
| MogDCN [10] | 49.21±4.99 | 2.44±0.74 | 0.76±0.63 | **0.996±0.003** | 6.840M | 47.48G | 45.14±5.41 | 3.19±1.45 | 1.75±1.66 | 0.980±0.019 | 7.444M | 47.48G |
| Fusformer [15] | 47.96±7.79 | 2.75±1.30 | 1.45±2.69 | 0.990±0.022 | 0.551M | 9.83G | 44.93±5.65 | 3.63±2.40 | 1.49±0.96 | 0.979±0.017 | 0.467M | 9.83G |
| DHIF [17] | 48.46±4.89 | 2.50±0.79 | 0.83±0.67 | **0.996±0.003** | 22.462M | 54.27G | 45.00±4.13 | 3.70±1.68 | 1.32±0.61 | 0.983±0.011 | 22.462M | 54.27G |
| PSRT [7] | 47.86±7.53 | 2.73±0.80 | 1.52±3.02 | 0.994±0.005 | 0.247M | 1.14G | 45.10±4.06 | 2.90±0.84 | 1.37±0.84 | 0.985±0.009 | 0.247M | 1.14G |
| 3DT-Net [25] | 49.41±5.83 | **2.26±0.66** | 0.83±1.07 | **0.996±0.003** | 68.07G | | 44.41±5.38 | 2.93±0.88 | 1.55±0.89 | 0.983±0.010 | 3.464M | 68.07G |
| DSPNet [35] | 49.18±4.84 | 2.57±0.79 | 0.75±0.62 | **0.996±0.003** | 6.064M | 6.81G | 45.84±3.62 | 2.97±0.75 | 1.33±0.64 | 0.984±0.010 | 6.064M | 6.81G |
| MIMO-SST [11] | 48.31±5.04 | 2.88±0.86 | 0.89±0.79 | 0.995±0.004 | 4.983M | 1.58G | 46.59±3.34 | 2.91±0.75 | 2.29±1.03 | 0.985±0.009 | 4.983M | 1.58G |
| FeINFN(Ours) | **50.32± 5.17** | 2.33±0.75 | **0.67±0.60** | **0.996±0.003** | 3.165M | 10.53G | **46.89±3.59** | **2.78±0.73** | **1.16±0.57** | **0.986±0.009** | 3.165M | 10.53G |

Table 6: The four average QIs and the corresponding parameters on the 11 testing images from the CAVE dataset simulating a scaling factor of 8. $\mathcal{S}$ & $\mathcal{F}$ means the domain difference.

| $\mathcal{S}$ | $\mathcal{F}$ | PSNR(↑) | SAM(↓) | ERGAS(↓) | SSIM(↑) |
|---|---|---|---|---|---|
| ✓ | ✗ | 50.01±5.33 | 2.36±0.72 | 0.71±0.68 | 0.996±0.0031 |
| ✗ | ✓ | 47.60±4.18 | 4.24±1.69 | 0.88±0.61 | 0.993±0.0041 |
| ✓ | ✓ | **50.32±5.17** | **2.33±0.75** | **0.67±0.60** | **0.996±0.0028** |

## A.11 Ablation Study: The Effectiveness of Decoder with Complex Gabor Wavelet Activation

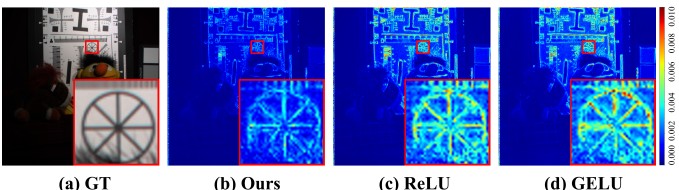

| (a) GT | (b) Ours | (c) ReLU | (d) GELU |
|---|---|---|---|

Figure 11: Error map for fusing an image with edges.

The Gabor wavelet activation demonstrates high representational power for visual signals, as depicted in Fig. 11. Compared to other activation functions, we observe that models utilizing the Gabor wavelet function exhibit lower error and spatial compactness.

