# OpenReview forum: "Fourier-enhanced Implicit Neural Fusion Network for Multispectral and Hyperspectral Image Fusion"
_NeurIPS.cc/2024/Conference — NeurIPS 2024 poster_

### Official Review · Reviewer_2Qbn · 2024-07-04

**Soundness:** 3
**Presentation:** 3
**Contribution:** 3
**Rating:** 6
**Confidence:** 4

**Summary:**

The paper provides a novel INR-based fusion framework tailored for the MHIF task, effectively capturing high-frequency details and global information through innovative architectural components. The key contributions are transforming the latent features into the frequency domain to enhance high-frequency information and introducing a complex Gabor wavelet activation function to strengthen feature representation and fusion.  Experiments on two MHIF datasets demonstrate the state-of-the-art performance achieved by the proposed method, outperforming existing approaches both visually and quantitatively.

**Strengths:**

Originality: This paper provides a novel solution to the problem of MLP-ReLU losing high-frequency information by leveraging domain transformation. The originality is high.
Quality: The viewpoints presented in the paper are supported by theoretical proofs and corresponding experimental evidence.
Clarity: The paper is well-organized and the content is clear and easy to read.
Significance: The proposed solution could have a significant impact on the field of computer vision.

**Weaknesses:**

- Complexity: Some parts of the methodology, particularly the theoretical proofs and implementation details, are complex and may be challenging for readers to fully grasp without additional explanation.
- Quality Metrics: The paper does not seem to provide the meaning and calculation methods of the four metrics (PSNR, SAM, ERGAS, SSIM).
- Experimental Section: Besides the two benchmark MHIF datasets used in the experiments, how does the proposed method perform in other image fusion tasks or datasets? It would be better if there were experiments on other datasets, but it's not essential.

**Questions:**

1. How does Fourier transformation within the INR framework specifically contribute to observed performance improvements? What are the practical benefits of frequency-domain features?
2. After reviewing the proof of Theorem 1, I don't understand why finding the optimal bandwidth leads to better representation. Could you provide further explanation, especially regarding Fig. 5.
3. For the PSNR curve in Figure 7, on which dataset was this experiment conducted? Was it on the training set, validation set, or test set? The paper does not explicitly specify this.
4. Regarding Fig. 6, were the errors normalized? The more the error maps lean towards red, the greater the difference between the pixel values and the ground truth. Is my understanding correct? The differences between the close-up images seem very small to the naked eye, and I can't discern any differences between the images in the first and third rows.

**Limitations:**

The authors have included a discussion on the limitations of their work in Section A.1, covering the data simulation, and potential societal impacts.

---

> ### Author Rebuttal · Authors · 2024-08-04
>
> **A1.** Thank you for your detailed review. We have revisited the sections that were considered complex and have worked to simplify the descriptions. For example, we have included foundational knowledge on INR and Fourier Transforms directly in the manuscript. Although the theoretical aspects involve signal processing concepts, such as Fourier Transforms, the underlying principles of the model’s design are not overly complex, and the implementation is relatively straightforward.
>
> **A2.** We have added detailed descriptions and calculation methods for each metric in the revised manuscript. Specifically:
> > **PSNR** evaluates the spatial quality of each band in the reconstructed HR-HSI. It is calculated as follows:
> $\text{PSNR}(\mathbf{I}, \tilde{\mathbf{I}}) = \frac{1}{B} \sum_{i=1}^{B} \text{PSNR}(\mathbf{I}^i, \tilde{\mathbf{I}}^i), $
> here, $\mathbf{I}^i \in \mathbb{R}^{H \times W}$ and $\tilde{\mathbf{I}}^i \in \mathbb{R}^{H \times W}$ represent the $i$-th band. The PSNR function is defined as: $\text{PSNR}(\mathbf{I}^i, \tilde{\mathbf{I}}^i) = 20 \cdot \log_{10} \left( \frac{\max(\mathbf{I}^i)}{\sqrt{\text{MSE}(\mathbf{I}^i, \tilde{\mathbf{I}}^i)}} \right)$.
>
> > **SAM** measures the spectral distortion of each hyperspectral pixel in the reconstructed HR-HSI. It is given by: $\text{SAM}(\mathbf{I}, \tilde{\mathbf{I}}) = \frac{1}{HW} \sum_{i=1}^{HW} \cos^{-1} \left( \frac{\mathbf{I}_i^T \tilde{\mathbf{I}}_i}{||\mathbf{I}_i||_2 ||\tilde{\mathbf{I}}_i||_2} \right)$, $||\cdot||_2$ is the $\ell_2$ norm, and $\cdot^T$ denotes the transpose.
>
> > **ERGAS** measures the global statistical quality of the reconstructed HR-HSI. It is formulated as: $\text{ERGAS}(\mathbf{I}, \tilde{\mathbf{I}}) = \frac{100}{c} \sqrt{\frac{1}{B} \sum_{i=1}^{B} \frac{\text{MSE}(\mathbf{I}^i, \tilde{\mathbf{I}}^i)}{\mu_{\tilde{\mathbf{I}}^i}^2}}, $ where $c$ is the scaling factor, and $\mu_{\tilde{\mathbf{I}}^i}^2$ is the square of the mean value of $\tilde{\mathbf{I}}^i$.
>
> >**SSIM** is used to assess the structural differences between GT and the reconstructed HR-HSI. It is defined as: $\text{SSIM}(\mathbf{I}, \tilde{\mathbf{I}}) = \frac{1}{B} \sum_{i=1}^{B} \frac{(2\mu_{I^i}\mu_{\tilde{I}^i} + C_1)(2\sigma_{I^i\tilde{I}^i} + C_2)}{(\mu_{I^i}^2 + \mu_{\tilde{I}^i}^2 + C_1)(\sigma_{I^i}^2 + \sigma_{\tilde{I}^i}^2 + C_2)}$. $\mu_{I^i}$ and $\mu_{\tilde{I}^i}$ represent the mean values of $I^i$ and $\tilde{I}^i$, while $\sigma_{I^i}^2$ and $\sigma_{\tilde{I}^i}^2$ denote their variances.
>
> **A3.**  In addition to the two benchmark MHIF datasets used in our experiments, we have also tested our method on outdoor remote sensing hyperspectral image datasets, specifically the Pavia Centre dataset and the Chikusei dataset. The detailed experimental results are presented in the response to Reviewer EFSC's Question 6 for specific details.
>
> **A4.**  **How:** (1) Neural networks, including MLPs, typically exhibit a bias toward learning low-frequency components more readily than high-frequency ones. This spectral bias can limit the network's ability to capture fine details. Transforming latent codes into the Fourier domain allows the framework to improve high-frequency injection and accurate image reconstruction. (2) In the frequency domain, a single frequency point corresponds to multiple pixels in the spatial domain. This relationship effectively enlarges the receptive field of the INR when operating in the frequency domain. Consequently, the network can capture more extensive contextual information and spatial correlations, which enhances its ability to model complex spatial dependencies and results in improved image quality and detail preservation.
>
> **What:** Frequency-domain features effectively preserve high-frequency details, crucial for tasks like super-resolution and texture synthesis. (1) The Fourier transform captures global patterns and structures efficiently, aiding tasks requiring holistic data understanding. (2) Integrating spatial and frequency features using an SFID and Gabor wavelet activation improves image reconstruction accuracy and robustness.
>
> **A5.** In the proof of Theorem 1, we primarily discuss the time-frequency tightness property of the Gabor wavelet activation function.
>
> **The Gabor wavelet activation**: Please refer to the response to Reviewer EFSC's Question 5 for specific details.
>
> **Regarding Fig. 5**, (c), (d), and (e) compare the frequency characteristics of fusion images generated by different activation functions with the GT images. The model using the complex wavelet activation function demonstrates better frequency fitting compared to the model using ReLU. This is evident in the comparison between (c) and (d), where the frequency distributions in the optimal bandwidth region are closer. In contrast, the frequency distributions in (c) and (e) show significant differences. By selecting the optimal bandwidth, we ensure that the Gabor wavelet achieves the best performance in processing time-frequency information, thereby enhancing the overall performance of the network.
>
> **A6.**  In Fig. 7 of our manuscript, the PSNR curve represents the performances **on the validation set**, plotted using the checkpoints saved per 5k iterations. As shown at the 80k-th iteration, the image exhibits high details, and loss on the testing set is lower, proving the effectiveness of the proposed IFF module.
>
> **A7.** You are correct.  **The error maps in Figure 6 are indeed normalized** to visually highlight the differences between the ground truth (GT) and predicted images. The performance of different methods is quite similar in visual observation since we **only select three bands from multiple bands to display as RGB images**, which cannot show the differences between all the bands very clearly. In addition, **the multispectral image has 16 bits of data**, thus, the different methods of visual observation are even small. Usually, we demonstrated the difference by numerical experiment and spectral vectors.

---

### Official Review · Reviewer_EFSC · 2024-07-06

**Soundness:** 4
**Presentation:** 3
**Contribution:** 4
**Rating:** 7
**Confidence:** 5

**Summary:**

The paper introduces a novel Fourier-enhanced Implicit Neural Fusion Network .The core innovation lies in the integration of Fourier transformations within an Implicit Neural Representation framework to address the loss of high-frequency information—a common limitation in existing INR approaches. The effectiveness of FeINFN is demonstrated through experiments on two benchmark MHIF datasets, showing superior performance in terms of PSNR and other metrics compared to state-of-the-art methods.

**Strengths:**

The paper introduces novel components to the INR framework, specifically tailored for MHIF tasks, a creative combination of existing ideas. The proposed modules are theoretically feasible. It provides clear and understandable graphical explanations for complex theories, such as Fig. 5. The paper is generally well-written, with clear explanations of complex concepts and a logical flow of ideas. The method proposed in the article performs optimally on two publicly available datasets, which should contribute significantly to the MHIF field.

**Weaknesses:**

1. Some sections of the paper are overly technical and dense, which might be challenging for readers unfamiliar with the specific domain of Fourier transforms and neural networks.

2. In Table 3, the description of S and F in the last row is incorrect and should be indicated with two check marks. This seems to be an editing error.

3. In the ablation experiments section of the paper, tables 2, 3, and 4 do not provide the corresponding parameter quantities. If available, could you please provide them? Ablation experiments need to compare models under similar parameter conditions.

**Questions:**

1. Network Complexity: The introduction of Spa-Fre IFF and SFID seems to increase the model complexity. Could the authors comment on the computational efficiency of their proposed method compared to the baselines?

2. In the explanation of the compound Gabor wavelet activation function in the article, Theorem 1 mentions optimal bandwidth. What does this refer to, and how does it affect the activation of features?

3. The abstract of the paper claims that the proposed model achieved state-of-the-art performance on two datasets. For the Harvard x4 experiments, how can the fact that the SAM and SSIM metrics did not reach SOTA be explained?

4. “frequency point may correspond to multiple pixels at different positions in the spatial domain”， Why does transforming into the Fourier domain amplify the frequency domain? I'm a bit unclear on this point.

**Limitations:**

The authors have provided reasonable Limitations and Broader Impact.

---

> ### Author Rebuttal · Authors · 2024-08-04
>
> **A1.** Thank you for your comments. We have revisited the sections that were considered overly technical and dense and have made efforts to simplify the descriptions. For instance, **we have included preliminary on INR and Fourier Transforms directly in the main text. The background on these topics is now presented in Sections 3.1 and Supplementary A.4.** Although the theory involves signal processing concepts, the principles underlying the model's design are not overly complex, and its implementation is relatively straightforward.
>
> **A2.** We have corrected the description of S and F in the last row of Table 3 to include two check marks as intended.
>
> **A3.**  We have included the parameter quantities for Tab. 2 and Tab. 3 in the ablation experiments section. As Tab. 4 evaluates the effectiveness of different activation functions, changing the activation function alone does not affect the parameter quantities, so no modifications were made to Tab. 4. We will update the relevant sections of the manuscript to reflect these changes.
> |Methods|Params|
> |:-|-:|
> |Bilinear|3.119M|
> |Bicubic|3.119M|
> |Pixel Shuffle|3.057M|
> |Our|3.165M|
>
> |$\mathcal{S}$/$\mathcal{F}$|Params|
> |:-|-:|
> |$\checkmark$/✘|2.869M|
> |✘/$\checkmark$|2.940M|
> |$\checkmark$/$\checkmark$|3.165M|
>
> **A4.** Comparing the computational efficiency of different methods is indeed important. To address this, we have added the FLOPs for each method and will update the final version of our paper to include a detailed comparison and analysis. We provide both the time and spatial complexities of our proposed model. Please refer to the response to Reviewer EUje's Question 3 for specific details.
>
> **A5.** Thank you for your insightful question.
>
> **Explanation of Optimal Bandwidth in Theorem 1**: Theorem 1 addresses the concept of optimal bandwidth in the context of the compound Gabor wavelet activation function. The optimal bandwidth refers to the ideal range of frequencies that the Gabor wavelet function can effectively capture. This balance is crucial because it ensures that the Gabor wavelet activation can efficiently represent both low and high-frequency components of the signal.
>
> **How Optimal Bandwidth Affects Feature Activation**: (1) The Gabor wavelet activation is designed to have time-frequency tightness, meaning it maintains high precision in both the time and frequency domains. This is governed by the uncertainty principle, $|\omega_0| \cdot \nu_0 \geq \frac{1}{4\pi}$ , which ensures a balanced representation. (2)  The optimal bandwidth allows the activation function to capture detailed information, improving the network's ability to reconstruct and represent high-frequency details. This leads to more accurate feature activation, particularly for detailed textures and fine-grained information. (3) During training, the network automatically learns the optimal bandwidths, resulting in precise and effective feature activations. This property of the Gabor wavelet activation ensures that the features align closely with the ground truth, enhancing the overall performance of the model. (4) The time-frequency tightness and optimal bandwidth of the Gabor wavelet activation function lead to better performance metrics, compared to other activation functions like ReLU and GELU. This results in enhanced fusion quality and robust feature learning within the Spatial-Frequency Interactive Decoder.
>
> **A6.** For the Harvard x4 experiments, while the SAM and SSIM metrics did not reach SOTA, our method's performance is very close to that of 3DT-Net, with only a difference of 0.06 in SAM and 0.001 in SSIM. Additionally, our FeINFN model achieved SOTA performance in Harvard x8, demonstrating even more significant improvements.
>
> Furthermore, we have conducted experiments on remote sensing hyperspectral image datasets, including the Pavia Centre dataset and the Chikusei dataset. These experiments, compared with several leading traditional and deep learning methods, also show that our model achieves SOTA results. Detailed results are presented in the table below:
> |Methods|Chikusei $\times 4$ (PSNR)|Chikusei $\times 4$ (SSIM) |Pavia $\times 4$ (PSNR)|Pavia $\times 4$ (SSIM)|
> |-|-|-|-|-|
> |ResTFNet|42.33|0.950|33.56|0.885|
> |SSRNet|42.36|0.951|33.20|0.876|
> |HSRNet|42.01|0.947|32.17|0.867|
> |MogDCN|42.21|0.936|33.84|0.889|
> |Fusformer|43.37|0.959|35.31|0.924|
> |DHIF|43.69|0.960|35.30|0.924|
> |PSRT|43.48|0.961|34.86|0.916|
> |3DT-Net|43.53|**0.963**|35.10|**0.927**|
> |DSPNet|43.55|0.960|35.47|0.927|
> |MIMO-SST|43.36|0.958|35.37|0.922|
> |**Proposed**|**43.88**|**0.963**|**35.51**|**0.927**|
>
> > Due to the rebuttal length limitation, the full metrics will be provided in the discussion stage.
>
> **A7.**  In the spatial domain, a single frequency point may correspond to multiple pixels at different positions. This is because the Fourier transform decomposes an image into a sum of sinusoidal functions of varying frequencies. Each frequency component has a global influence across the entire image, meaning that a specific frequency can affect multiple spatial locations.
>
> This amplification effect occurs due to the following reasons: (1) The Fourier transform separates the image into its constituent frequency components, making certain features more distinguishable in the frequency domain than in the spatial domain. This separation aids in identifying and manipulating specific frequency characteristics. (2) Convolution operations in the frequency domain may affect the entire spatial domain. A single point in the frequency domain corresponds to multiple points in the spatial domain, resulting in a global impact that makes certain features more prominent. (3) Operations in the Fourier domain can enlarge the receptive field of corresponding spatial domain operations. This means localized features in the frequency domain can be represented more globally in the spatial domain, leading to an apparent amplification of certain features.

---

> > ### Author Response · Authors · 2024-08-08
> > **Supplement about the complete version of  tables in rebuttal**
> >
> > **A3.**  Tab 2,3 updates as shown below：
> > |Methods|PSNR ($\uparrow$)|SAM ($\downarrow$) |ERGAS ($\downarrow$) |SSIM ($\uparrow$)|Params|
> > |:-:|:-:|:-:|:-:|:-:|:-:|
> > |Bilinear|52.23$\pm$4.40|1.92$\pm$0.60|1.03$\pm$0.86|0.997$\pm$0.0021|3.119M|
> > |Bicubic|52.22$\pm$4.31|1.95$\pm$0.61|1.02$\pm$0.82|0.997$\pm$0.0021|3.119M|
> > |Pixel Shuffle|52.26$\pm$4.37|**1.90$\pm$0.59**|1.02$\pm$0.85|0.997$\pm$0.0022|3.057M|
> > |Our|**52.47$\pm$4.10**|1.91$\pm$0.59|**0.98$\pm$0.74**|**0.998$\pm$0.0015**|3.165M|
> >
> > |$\mathcal{S}$/$\mathcal{F}$|PSNR ($\uparrow$)|SAM ($\downarrow$)|ERGAS ($\downarrow$)|SSIM ($\uparrow$)|Params|
> > |:-:|:-:|:-:|:-:|:-:|:-:|
> > |$\checkmark$/✘|52.11$\pm$4.22|1.95$\pm$0.59|1.04$\pm$0.82|0.998$\pm$0.0017|2.869M|
> > |✘/$\checkmark$|47.86$\pm$3.42|3.49$\pm$1.30|1.67$\pm$1.13|0.995$\pm$0.0020|2.940M|
> > |$\checkmark$/$\checkmark$|**52.47$\pm$4.10**|**1.91$\pm$0.59**|**0.98$\pm$0.74**|**0.998$\pm$0.0015**|3.165M|
> >
> >
> >
> > **A6.**  Detailed results on the Chikusei dataset and Pavia Center dataset are as follows:
> > | |Chikusei $\times 4$| | | | |Pavia $\times 4$| | | | |
> > |:-:|:-:|:-:|:-:|:-:|:-:|:-:|:-:|:-:|:-:|:-:|
> > |**Methods**|**PSNR ($\uparrow$)**|**SAM ($\downarrow$)**|**ERGAS ($\downarrow$)**|**SSIM ($\uparrow$)**|**Params** |**PSNR ($\uparrow$)**|**SAM ($\downarrow$)**|**ERGAS ($\downarrow$)**|**SSIM ($\uparrow$)**|**Params**|
> > |Bicubic|33.35|4.00|7.65|0.815|-|26.65|7.07|8.46|0.614|-|
> > |CSTF-FUS|35.40|5.40|7.88|0.844|-|30.93|11.08|5.74|0.791|-|
> > |LTTR|37.86|3.68|6.27|0.917|-|31.15|6.58|5.50|0.801|-|
> > |LTMR|41.21|2.98|4.84|0.950|-|32.33|6.35|5.10|0.820|-|
> > |IR-TenSR|36.00|5.12|7.86|0.868|-|30.87|6.81|5.82|0.783|-|
> > |ResTFNet|42.33|2.48|3.93|0.950|2.471M|33.56|4.99|4.16|0.885|2.448M|
> > |SSRNet|42.36|2.35|3.92|0.951|0.446M|33.20|5.36|4.36|0.876|0.282M|
> > |HSRNet|42.01|2.33|3.95|0.947|0.633M|32.17|5.60|4.60|0.867|2.061M|
> > |MogDCN|42.21|2.27|3.76|0.936|6.840M|33.84|4.61|4.07|0.889|7.202M|
> > |Fusformer|43.37|2.03|3.49|0.959|0.504M|35.31|4.33|3.37|0.924|0.539M|
> > |DHIF|43.69|1.94|3.33|0.960|22.462M|35.30|4.36|3.35|0.924|38.785M|
> > |PSRT|43.48|2.01|3.47|0.961|0.303M|34.86|4.47|3.54|0.916|0.288M|
> > |3DT-Net|43.53|2.03|3.46|**0.963**|3.464M|35.10|4.44|3.35|**0.927**|3.482M|
> > |DSPNet|43.55|2.03|3.44|0.960|6.138M|35.47|4.26|**3.30**|**0.927**|6.115M|
> > |BDT|43.25|2.09|3.44|0.955|3.263M|34.55|4.66|3.70|0.904|3.056M|
> > |MIMO-SST|43.36|2.09|3.48|0.958|4.983M|35.37|4.48|3.34|0.922|5.227M|
> > |**Proposed**|**43.88**|**1.91**|**3.30**|**0.963**|3.488M|**35.51**|**4.15**|**3.30**|**0.927**|3.399M|

---

> > ### Comment · Reviewer_EFSC · 2024-08-10
> > **Comments on the rebuttal**
> >
> > The response addressed all my comments well. Therefore, I'll raise the score to ACCEPT.

---

### Official Review · Reviewer_8xGR · 2024-07-11

**Soundness:** 4
**Presentation:** 3
**Contribution:** 4
**Rating:** 7
**Confidence:** 4

**Summary:**

The paper proposes a novel Fourier-enhanced Implicit Neural Fusion Network (FeINFN) specifically designed for the Multispectral and Hyperspectral Image Fusion task. The paper identifies the unique characteristics of the amplitude and phase of the latent codes in both HRHSI and LRHSI, and proposes to enhance high-frequency details and expand the receptive field from the Fourier domain. Additionally, the paper introduces a new decoder to achieve better interaction between spatial domain features and frequency domain features.

**Strengths:**

1. As mentioned in the Summary, the paper's observations on amplitude and phase are intriguing, and the proposed spatial-frequency domain fusion framework aligns with the paper's motivation.

2. The paper presents clear theoretical proofs for its claims, such as the advantage of complex Gabor wavelet activation in finding the optimal bandwidths.

3. The paper is well-written, with clear and organized content.

4. The experimental observations are interesting and can be considered to have a positive impact.

**Weaknesses:**

1.	What does 'current neural network-based methods are insensitive to high-frequency information' mean, and how does the proposed model demonstrate sensitivity to high-frequency information?
2.	As mentioned in Section 3.4, the Spatial Implicit Fusion Function utilizes frequency encoding in the spatial domain. Does the Frequency Implicit Fusion Function also utilize frequency encoding in the frequency domain?
3.	Should the vertical axis scales of (c), (d), and (e) in Figure 5 be consistent? In my understanding, at least (c) and (d) should be consistent to demonstrate that the ground truth (GT) frequencies and the frequencies captured using Gabor activation are similar.
4. This paper proposes a new fusion architecture based on INR. It would be even better if there were discussions and analyses regarding network complexity, such as comparisons in spatial complexity with Transformer methods mentioned in section I.

5. There are some editing errors: the double quotes are displayed incorrectly in lines 307 and 310 as “Fourier Domain" and “chart and stuffed toy".

6. In line 226, “I” should correspond to elsewhere in the article.

**Questions:**

See weaknesses

---

> ### Author Rebuttal · Authors · 2024-08-03
>
> **Q1.** What does 'current neural network-based methods are insensitive to high-frequency information' mean, and how does the proposed model demonstrate sensitivity to high-frequency information?
>
> **A1.** Thank you for your careful review. **(1)** High-frequency insensitivity is a common issue with MLPs, which are the primary architecture of INR. Leveraging this insight, we propose a method to enhance high-frequency information in the frequency domain. **(2)** Ablation experiments (i.e., Sec. 4.1 in the manuscript) verify that our model alleviates the inherent shortcomings in INR, namely high-frequency insensitivity. Fig. 7 of the manuscript demonstrates that our model exhibits faster convergence and overall higher efficiency during training stages and exhibits more high-frequency details visually at the same iteration.
>
> **Q2.** As mentioned in Section 3.4, the Spatial Implicit Fusion Function utilizes frequency encoding in the spatial domain. Does the Frequency Implicit Fusion Function also utilize frequency encoding in the frequency domain?
>
> **A2.** Thank you for your question. In the frequency domain, we do not use frequency encoding. We tested the inclusion of frequency encoding in the frequency domain, but it did not yield any performance improvement in our experiments. *Consider the non-performance improvement and introduce another computational overhead*, therefore, we choose not to incorporate it to avoid unnecessary complexity.
>
> **Q3.** Should the vertical axis scales of (c), (d), and (e) in Figure 5 be consistent? In my understanding, at least (c) and (d) should be consistent to demonstrate that the ground truth (GT) frequencies and the frequencies captured using Gabor activation are similar.
>
> **A3.** Thank you for pointing this out. You are totally correct in your understanding. The vertical axis scales in subfigures (c) and (d) of Fig. 5 are indeed consistent; the scale of 1500 is simply not displayed in (c).
> Subfigures (c), (d), and (e) in Fig. 5 compares the frequency characteristics of fusion images generated by different activation functions with the ground truth (GT) images. The model using the complex wavelet activation function demonstrates better frequency fitting compared to the model using ReLU. This is evident in the comparison between (c) and (d), where the frequency distributions in the optimal bandwidth region are closer. In contrast, the frequency distributions in (c) and (e) show significant differences.
>
> **Q4.** This paper proposes a new fusion architecture based on INR. It would be even better if there were discussions and analyses regarding network complexity, such as comparisons in spatial complexity with Transformer methods mentioned in section I.
>
> **A4.** Thank you for your insightful suggestion. We have calculated both the computational and spatial complexities of our proposed model.
> **The time complexity** of our model is $4D^2L\approx \mathcal O(L)$ (because usually $L\gg D$), where $D$ represents the number of channels in the input image and $L$ represents the image dimensions (*i.e.,* $H\times W$). In comparison, Transformer-based methods have a time complexity of $3D^2L+2DL^2\approx \mathcal O(L^2)$. Our model's time complexity is linear to sequence length, whereas other Transformer methods (*e.g.,* PSRT) scale with $\mathcal O(L^2)$.
> Additionally, **the spatial complexity** (*a.k.a,* memory consumption) of the attention operator in Transformers is $\mathcal O(L^2)$, which can lead to out-of-memory issues with high-resolution images. In contrast, the spatial complexity of the MLP in our INR-based model does not grow quadratically for the image size. This means that our model can handle larger images without encountering the same memory constraints as Transformer methods.
>
> **Q5.** There are some editing errors: the double quotes are displayed incorrectly in lines 307 and 310 as “Fourier Domain" and “chart and stuffed toy".
>
> **A5.** Thank you for pointing out the editing errors. We apologize for the oversight and have corrected the double quotes as indicated in lines 307 and 310.
>
> **Q6.** In line 226, “I” should correspond to elsewhere in the article.
>
> **A6.** Thank you for bringing this to our attention. We have corrected the reference to “I” in line 226 to ensure it corresponds appropriately with other parts of the article.

---

> > ### Comment · Reviewer_8xGR · 2024-08-11
> >
> > The response addressed most of my concerns. Thus, I will raise the score to accept.

---

### Official Review · Reviewer_EUje · 2024-07-11

**Soundness:** 3
**Presentation:** 3
**Contribution:** 3
**Rating:** 6
**Confidence:** 5

**Summary:**

This paper proposes a quite interesting hyperspectral and multispectral image fusion framework via implicit representation. Moreover, the authors introduce the Fourier transformation to decouple the amplitude and phase domain.

**Strengths:**

1.	Introducing implicit model into the task of hyperspectral multispectral fusion is quite interesting.

2.	A novel Gabor wavelet activation function is proposed. Theoratical analysis of Gabor activation is also given;

3.	The decoupling process of spatial-spectral domain is reasoable for hyperspectral image;

4.	The performance of the proposed method is competitative with SOTA methods; and

5.	Paper is easy to follow, figure is clear to vlidate the motivation of designing.

**Weaknesses:**

1.	The proposed Fourier decomposition seems to only result in very limited performance improvement. The results from table 3 and figure 7 seem not consistent.
2.	Actually, such implicit representation is more plausible to work in a self-regularized overfitting manner than current train-test setting. Authors can make further design or give a unsupervised version.
3.	Authors should also compare the computational cost of different methods.
4.	Authors should also evaluate the performance with different SR ratios.

**Questions:**

Improvement of Fourier design.
Unsupervised training manner.
Computational cost.
Different SR ratios.
As the weakness Sec.

**Limitations:**

Please refer to weakness sec.

---

> ### Author Rebuttal · Authors · 2024-08-04
>
> **Q1.** Improvement of Fourier design. The results from table 3 and figure 7 seem not consistent.
>
> **A1.** Thank you for your insightful comments. We apologize for any confusion caused by our unclear presentation. To clarify:
>
> - **Inconsistencies Between Tab. 3 and Fig. 7:** The results in Tab. 3 are derived from the *test set*, while Fig. 7 displays the performance on the *validation set* (since we can not access test set when training and validating).
> - **Limited Performance Improvement:** While the proposed Fourier decomposition may offer limited performance improvement, **our primary motivation extends beyond simply enhancing the results. Our goal was to address inherent issues within Implicit Neural Representations (INRs) as state in Sect. 2(Motivation)**. Specifically, as illustrated in Fig. 7, incorporating the Fourier domain for implicit representation accelerates the network's convergence compared to using only the spatial domain. This demonstrates that the network can focus more on high-frequency information in the early stages of training, mitigating the spectral bias introduced by MLP-ReLU architectures.
>
> **Q2.**  Unsupervised training manner.
>
> **A2.**  Thank you for your valuable suggestion. Inspired by your feedback, we propose an unsupervised approach that leverages the self-regularized overfitting manner without relying on high-resolution ground-truth images to fuse an image.
>
>  Specifically, **we select a LRHSI-HRMSI pair then downsample the pair by a fixed factor (e.g. 4). Then the downsampled pair is fed into the proposed INR network and produce the fused image. The self-regularized L1 loss is used to optimize the model, where the original LRHSI serves as the unsupervised GT.** This setup allows us to train in a more self-regularized, unsupervised manner. This training method does not conflict with our current training method. And our training diagram can enable our model to fuse images on different test pairs, not just on one single image. We appreciate your insightful suggestion and believe this approach could provide a promising direction for future work.
>
> **Q3.** Computational cost.
>
> **A3.** Thank you for your valuable suggestion. Comparing the computational cost of different methods is indeed important. To address this, we have added the FLOPs for each method and will update the final version of our paper to include a detailed comparison and analysis.
>
> We provide both the time and spatial complexities of our proposed model below:
> - **The time complexity** of our model is $4D^2L\approx \mathcal O(L)$ (because usually $L\gg D$), where $D$ represents the number of channels in the input image and $L$ represents the image dimensions (*i.e.,* $H\times W$). In comparison, Transformer-based methods have a time complexity of $3D^2L+2DL^2\approx \mathcal O(L^2)$. Our model's time complexity is linear to sequence length, whereas other Transformer methods (*e.g.,* PSRT) scale with $\mathcal O(L^2)$.
> - **The spatial complexity** (*a.k.a,* memory consumption) of the attention operator in Transformers is $\mathcal O(L^2)$, which can lead to out-of-memory issues with high-resolution images. In contrast, the spatial complexity of the MLP in our INR-based model does not grow quadratically for the image size. This means that our model can handle larger images without encountering the same memory constraints as Transformer methods.
>
> The table below shows the number of paramters and FLOPs among our method and other methods.
>
> |Methods|CAVE $\times 8$ PSNR ($\uparrow$)|CAVE $\times 8$ SAM ($\downarrow$)|CAVE $\times 8$ ERGAS ($\downarrow$)|CAVE $\times 8$ SSIM ($\uparrow$)|Harvard $\times 8$ PSNR ($\uparrow$)|Harvard $\times 8$ SAM ($\downarrow$)|Harvard $\times 8$ ERGAS ($\downarrow$)| Harvard $\times 8$ SSIM ($\uparrow$)|Params|FLOPs|
> |-|-|-|-|-|-|-|-|-|-|-|
> |Bicubic|29.96|5.89|5.56|0.887|33.18|3.10|3.83|0.894|-|-|
> |CSTF-FUS|38.44|7.00|2.11|0.959|39.84|4.49|2.40|0.932|-|-|
> |LTTR|37.92|5.37|2.44|0.972|42.09|3.62|1.80|0.960|-|-|
> |LTMR|38.41|5.04|2.24|0.974|42.09|3.62|1.80|0.959|-|-|
> |IR-TenSR|36.79|12.87|2.68|0.944|40.04|5.40|4.75|0.958|-|-|
> |ResTFNet|43.77|3.49|1.38|0.992|43.50|3.53|1.74|0.979|2.387M|1.75G|
> |SSRNet|46.23|3.13|1.05|0.993|45.76|2.99|1.34|0.983|0.027M|0.11G|
> |HSRNet|46.69|2.91|0.93|0.994|44.02|3.64|1.49|0.980|1.09M|2.00G|
> |MogDCN|49.21|2.44|0.76|0.996|45.14|3.19|1.75|0.980|6.840M|47.48G|
> |Fusformer|47.96|2.75|1.45|0.990|44.93|3.63|1.49|0.979|0.504M|9.83G|
> |DHIF|48.46|2.50|0.83|0.996|45.00|3.70|1.32|0.983|22.462M|54.27G|
> |PSRT|47.86|2.73|1.52|0.994|45.10|2.90|1.37|0.985|0.247M|1.14G|
> |3DT-Net|49.41|**2.26**|0.83|0.996|44.41|2.93|1.55|0.983|3.464M|68.07G|
> |DSPNet|49.18|2.57|0.75|0.996|45.84|2.97|1.33|0.984|6.064M|6.81G|
> |MIMO-SST|48.31|2.88|0.89|0.995|46.59|2.91|2.29|0.985|4.983M|1.58G|
> |**Proposed**|**50.32**|2.33|**0.67**|**0.996**|**46.89**|**2.78**|**1.16**|**0.986**|3.165M|10.53G|
>
> **Q4.** Different SR ratios.
>
> **A4.** Thank you for your valuable comment. We have conducted additional fusion experiments on the CAVE and Harvard datasets with an SR ratio of 8. The results of these experiments can be found in supplementary A. 8. Furthermore, we have included a comparative visual illustration in Fig. 1, showcasing the performances of our method at SR ratios of x4 and x8 against other approaches. The results indicate that our model performs better under different fusion ratios, which is attributed to the powerful continuous representation capability of INR.

---

> > ### Comment · Reviewer_EUje · 2024-08-08
> >
> > Thanks for the responses. My concerns are resolved. I'll raise the score to WA.

---

### Decision · Program_Chairs · 2024-09-25

**Decision:**

Accept (poster)

**Comment:**

The paper proposes a Fourier-enhanced implicit neural fusion network specifically designed for the multispectral and hyperspectral image fusion task. The paper identifies the unique characteristics of the amplitude and phase of the latent codes in the high-frequency and low- frequency components, and proposes to enhance high-frequency details and expand the receptive field from the Fourier domain.  Extensive experiments are theoretical analysis are provided in the submission and the rebuttal for validation. After the rebuttal, all the reviewers give positive recommendation for this submission.